# Plant–Soil Interactions Shape Arbuscular Mycorrhizal Fungal Diversity and Functionality in Eastern Tibetan Meadows

**DOI:** 10.3390/jof11050337

**Published:** 2025-04-25

**Authors:** Shihu Zhang, Zhengying Yang, Xuechun Yang, Xiaoyu Ma, Qun Ma, Miaojun Ma, Jiajia Zhang

**Affiliations:** 1College of Life Science, Northwest Normal University, Lanzhou 730070, China; m17393161590@163.com (Z.Y.); yxc15693831664@163.com (X.Y.); mxy12282025@163.com (X.M.); 13565700056@163.com (Q.M.); 2College of Ecology, Lanzhou University, Lanzhou 730000, China; mjma@lzu.edu.cn; 3School of Life Sciences, Institute of Life Science and Green Development, Hebei University, Baoding 071002, China; zhangjj@henu.edu.cn

**Keywords:** symbiotic fungi, fungal community, plant functional traits, soil properties, alpine meadow

## Abstract

Arbuscular mycorrhizal (AM) fungi occur in the interface between soils and plants. Yet, the impacts of the plant community functional composition and soil properties on AM fungal communities remain poorly understood in the face of ongoing climate change. Here, we investigated the AM fungal community in alpine meadow habitats of the Tibetan Plateau by linking fungal species richness to plant community functional composition and soil parameters at three latitudinal sites. High-throughput sequencing of the AM fungal small subunit rRNA gene was performed to characterize fungal communities. We found that AM fungal diversity and plant functional diversity, as well as the contents of soil nutrients, were significantly higher in the southernmost site, Hongyuan (HY). Total soil nitrogen and soil-available phosphorus explained the variation in AM fungal diversity, while AM fungal biomass was best predicted by the plant community-weighed mean nitrogen:phosphorus ratio (CWM-N:P). *Glomus* species preferentially occurred in the northernmost site of Hezuo (HZ). Distance-based redundancy analysis (db-RDA) revealed that AM fungal community structure was influenced by not only CWM-N:P but also by plant community-weighed mean photosynthetic rate (CWM-Pn), soil total carbon, and plant community functional dispersion (FDis). We conclude that plant traits and soil properties are crucial for nutrient–carbon (C) exchange, as fungal symbionts may shape AM communities in this vast alpine meadow ecosystem. Our findings provide timely insight into AM fungal community assembly from the perspective of nutrient–C exchange dynamics in the Tibetan Plateau’s alpine meadow habitats.

## 1. Introduction

In recent decades, human-induced climate change has contributed to altered precipitation patterns and hydrological cycles [1,2]. In particular, the magnitude, frequency, and duration of heavy precipitation events are being enhanced significantly by greater warming at global, continental, and regional scales [3,4]. Grassland ecosystems are an integral terrestrial component and are especially sensitive to precipitation changes since they are naturally limited by soil water availability [5]. Nevertheless, reduced rainfall, such as that leading to drought, also alters the composition and structure of grassland plant communities, further affecting ecosystem functioning and resiliency [6,7,8]. Plant community functional composition, whether defined by single traits (i.e., community-weighed trait means, CWMs) and/or multiple traits approach (i.e., trait distribution within communities), has been linked to the responses of grassland ecosystems to altered precipitation regimes [9,10,11]. High levels of functional trait diversity can often bolster overall resource utilization via complementarity dynamics among species, thereby conferring greater resilience to plant communities facing environmental changes [12,13]. Moreover, certain plant traits can mediate how grassland communities impact soil moisture [14]. For instance, a tall canopy limits the amount of light radiation reaching the ground surface, leading to more soil water being retained. Although plant traits are thought to play an essential role in soil functions, the linkages between plant community functional composition and the soil microbial community are far from fully understood [15].

Arbuscular mycorrhizal (AM) fungi (Glomeromycota) can form mycorrhizal symbioses with the roots of about 72% of terrestrial plant species [16], which constitute at least 55% of terrestrial vegetation’s cover [17] and 63% of its net primary productivity [18]. While AM fungi are widely distributed in all global regions, they are most prevalent in low and mid-latitude forest and grassland ecosystems, where they figure prominently in plants’ uptake of mineral nutrients and soil water provisioning [19,20,21]. Since AM fungi can establish mutualistic associations with most land plants, they link the aboveground and belowground parts of an ecosystem through complex mineral–nutrient–carbon exchanges, and this applies to most grasslands and their species [22,23]. In addition to directly affecting plants’ photosynthetic intensity by influencing their carbon’s source–sink relationship, mycorrhizal symbiosis can augment the shift to a belowground carbon sink from plants; this process reduces photosynthate accumulation in plants, inducing them to increase their photosynthetic rate to compensate for the carbon transferred [24,25]. For example, Gavito et al. (2019) found that plant photosynthetic rates were reduced by 10% to 40% within a short period of disconnecting a portion of their AM fungal hyphae [26]. This strongly suggests that mycorrhizal symbiosis can modulate the source–sink relationship of photosynthesized carbon in plants, which in turn drives the intensity of photosynthesis. AM fungi can improve plant fitness through the provision of nutritional and non-nutritional benefits for their hosts [20,27]. Recently, there has been emerging evidence showing that AM hyphae can directly provide host plants with water via roots [28]. In return, their plant hosts provide AM fungi with carbohydrates for the formation, maintenance, and function of mycorrhizal structures. In tandem, AM fungi produce extensive hyphal networks in roots and soil, intimately linking plants and soils [29].

Several studies have demonstrated that changed precipitation could alter the diversity and composition of AM fungal communities [30,31,32,33,34]. For example, long-term drought increased the AM fungal colonization of roots yet decreased the extraradical mycorrhizal hyphae densities in soils [35]. In reanalyzing published data from grasslands in Northern America [36], Rillig (2004) found a strong positive relationship (*R*^2^ = 0.97) between AM fungal hyphae densities and precipitation levels [37]. The responses of AM fungal communities to precipitation changes are known to be associated with precipitation-induced changes in plant community composition [31,35]. In addition to that close association, a plant community’s functional composition has been linked to how it responds to changes in precipitation patterns [9,11].

Alpine grasslands of the Tibetan Plateau are very sensitive to precipitation changes [38,39]. Although the mean annual precipitation in this area has increased sharply over recent decades [40], significant warming of the Tibetan Plateau is believed to promote the growth of alpine grasslands, leading to an increase in atmospheric aridity, and thereby constraining vegetation growth [39]. Most plants in these alpine grasslands are colonized by AM fungi, which are generally abundant [41]. Thus, a better understanding of these mutualistic associations is imperative for informing alpine grassland management and conservation in the face of rapid climate change [40], since AM fungi play a key role in plant nutrient availability, productivity, and the biodiversity of vegetation communities [42]. In alpine meadows, where harsh environmental conditions limit plant productivity and nutrient cycling, AM fungi often serve as key biotic drivers of ecosystem functionality [43]. Understanding the diversity and distribution of AM fungal communities in such ecosystems has become increasingly important in the context of global climate change and biodiversity loss.

While previous studies have characterized AM fungal diversity across different ecosystems, the interactive effects of plant community composition, soil properties, and environmental gradients on AM fungal assemblages remain poorly understood in high-altitude meadows. In particular, the eastern edge of the Tibetan Plateau represents a global hotspot for both alpine biodiversity and climate vulnerability, yet the mechanisms governing belowground plant–microbe interactions in these regions are still underexplored.

Plant–soil–microbe feedbacks, especially those involving vascular plant communities and their associated mycospheres, are central to determining the structure and function of AM fungal communities. However, most research has focused on either plant composition or fungal diversity in isolation. A more integrative perspective that examines how specific vascular plant species, their richness, and functional traits influence AM fungal assemblages is needed. Moreover, the role of climate-related stressors (e.g., moisture regimes) in shaping co-distributed changes in both aboveground and belowground communities warrants further investigation.

In this study, we used plant community-level traits and soil properties associated with carbon (C) and nutrients to investigate AM fungal community responses in three alpine meadow sites, given the following considerations. Previous studies have demonstrated that precipitation-driven shifts in plant community functional composition can influence AM fungal communities [31,35]. Moreover, changes in precipitation patterns have been shown to affect plant community traits, which in turn govern community-level functional composition [9,11]. Additionally, plant-derived C is expected to underpin the fitness of AM fungi because they are asexual obligate biotrophs that rely exclusively on C from host plants to complete their life cycle [29]. The ability of AM fungi to improve the water and photosynthetic profiles of plants is well-documented for agricultural and horticultural plants [26,44,45]. In contrast, far less is known about the modulation of photosynthesis by AM fungi in relation to the induced C sink strength mechanism in grassland ecosystems.

We measured responses of AM fungal biomass and diversity, various soil properties, plant community functional diversity (functional richness—FRic, functional evenness—FEve, and functional dispersion—FDis), community-weighted trait means (CWM) of specific leaf area (SLA), leaf nitrogen concentration (LNC), leaf phosphorus concentration (LPC), the leaf nitrogen and phosphorus ratio (N:P), and photosynthetic rate (Pn) to test the following three hypotheses: (1) Higher levels of soil nutrients would augment fungal diversity; (2) AM fungal biomass would respond to plant traits such as photosynthetic rate and plant community functional dispersion; and (3) both plant traits and soil properties involved in nutrient-C exchange would shape AM fungal community structures in the alpine meadow ecosystem of the Tibetan Plateau.

This study will advance the current state-of-the-art by simultaneously assessing AM fungal communities, vascular plant functional composition, and key edaphic parameters across an elevation gradient. This integrative approach offers new insights into the drivers of belowground biodiversity and plant–soil–microbe interactions under climate-sensitive alpine conditions.

## 2. Materials and Methods

### 2.1. Site Selection

We selected three alpine meadow sites at the eastern edge of the Tibetan Plateau (Appendix A). The first site (‘HY’) was located at the Alpine Wetland Ecosystem Field National Observation and Science Research Station of Hongyuan County in Sichuan Province, China (32°48′ N, 102°33′ E; 3508 m a.s.l.), where the mean annual precipitation is 690 mm and the mean annual temperature is 0.9 °C, with a mean vegetation coverage of 93.81% [46]. The second site (‘MQ’) was located at the Gannan Grassland Ecosystem National Observation and Research Station of Maqu County in Gansu Province, China (33°38′ N, 101°53′ E; 3500 m a.s.l.), having a mean annual precipitation and temperature of 620 mm and 1.2 °C, respectively [47,48]. There, the mean vegetation cover was lower, at 85.23%, due to rodent activity in the site’s plant community. The third site (‘HZ’) was located at the Hezuo branch station of the Gannan Grassland Ecosystem National Observation and Research Station in Gansu Province, China (34°55′ N, 102°53′ E; 2960 m a.s.l.), with a mean annual precipitation and temperature of 560 mm and 2.0 °C, respectively [49], and a mean vegetation cover of 91.35%.

The plant communities are dominated by perennial herbaceous species in the families Compositae, Gramineae, Ranunculaceae, Lamiaceae, Leguminosae, and Cyperaceae, which can nevertheless attain a high degree of richness: more than 30 spp. can co-occur within a 50-cm × 50-cm quadrat at these three sites [50,51]. These alpine meadow soils all belong to the Cambisol type (FAO taxonomy) [52]. Taken together, the environmental conditions and vegetation types correspond to typical alpine meadows on the eastern Tibetan Plateau. All three sampling sites were moderately grazed by yaks from October to May (when serving as winter pasture), with yaks moved to higher elevations from May to mid-September (as summer pasture) [53]. Detailed information on the soil physicochemical properties and plant species of the alpine meadow sites can be found in Table 1 and Appendix A, respectively.

### 2.2. Field Investigation and Soil Sampling

To examine the plant communities, we randomly selected 10 plots (each 10 m × 10 m) within a 100-m × 200-m area at each site, ensuring a minimum distance of 2 m between plots. For each plant species, its cover within a given plot was visually quantified in mid-August 2020 using a 1-m × 1-m quadrat consisting of a grid formed by 100 equally distributed cells (each 10 cm × 10 cm; with intersection points 10 cm apart). Community cover was summed across the species present each quadrat, and the relative cover of each species was calculated as the cover of the individual species divided by the community cover. Plant traits were also surveyed, but only for the most common species in the quadrat (i.e., those with an accumulated abundance > 90%). After plant community sampling, five soil cores (topsoil layer: 0–20 cm depth) were randomly collected from each plot. These soil samples were homogenized to yield a single composite sample per plot and then sieved through a 2-mm mesh after removing any plant roots and stones. To avoid contamination between soil samplings, the used tools were alcohol-sterilized each time (with the ethanol burnt off). Each soil composite sample was divided into three sub-samples. The first subsample was immediately placed inside an ice box for transport to the laboratory, where it was stored at −80 °C until its analysis. The second subsample was used to determine the soil water content (SWC); specifically, the fresh soil was placed in an aluminum box, weighed, and then oven-dried at 120 °C for 48 h in the laboratory. The third subsample was air-dried and used to determine the soil pH, total soil carbon (TSC), total soil nitrogen (TSN), total soil phosphorus (TSP), soil available nitrogen (SAN), and soil available phosphorus (SAP).

### 2.3. Measurement of Plant Traits and Calculation of Functional Diversity

The in situ photosynthetic rates (Pn) of dominant species were measured in mid-August 2020 on sunny days between 9:30 and 11:30 using the LI-6400 Portable Photosynthetic System (LI-COR, Lincoln, NE, USA). For this, mature and fully expanded leaves of 49 dominant herbaceous across the species present in quadrats: 36 forbs, 5 legumes, 2 sedges, and 6 grasses, spanning 16 families (Appendix A) were selected and measured using at least 10 replicate plants from across the 100-m^2^ plot, under an ambient CO_2_ concentration of 400 µmol mol-1 at 20–25 °C. Photosynthetic photon flux density (PPFD) was set at 1500 µmol quanta m^−2^ s^−1^ by using a red–blue 6400-02B LED light source (LI-COR Inc., Lincoln, NE, USA). All selected leaves were allowed to acclimate to those conditions in the chamber before taking any measurements. Three repeated values were recorded per leaf per plant until all parameters remained stable; mean values per species were calculated for use in the formal analysis. Plant community functional composition was assessed by three functional diversity indices: FRic, FEve, and FDis, respectively [54,55]. CWM was calculated as the weighted average of species abundance based on population functional traits to obtain community-weighted trait means [56]. Pn was included in the calculation of plant community functional composition. After measuring the species’ photosynthetic rate, all leaves were collected to determine their SLA, LNC, LPC, and N:P. The fresh leaves were imaged by an Epson Perfection V850 Pro scanner (Seiko Epson Corp., Nagano, Japan). After determining the respective leaf area of the scanned samples using WinFOLIA software (version 2022, Regent Instruments Inc., Quebec City, QC, Canada), they were oven-dried at 75 °C for 48 h and weighed. Next, the same leaf samples were digested in a mixture of sulphuric acid and hydrogen peroxide, and their LNC and LPC were quantified with a SmartChem^®^ 140 Discrete Analyzer (WESTCO Scientific Instruments Inc., Milan, Italy).

### 2.4. Measurement of Soil Properties

Several soil parameters were measured according to standard protocols [47]. From each first subsample, 5 g of frozen soil was used to determine SAN (NH^4+^-N and NO^3−^-N) with the SmartChem^®^ 140 Discrete Analyzer (WESTCO Scientific Instruments Inc., Milan, Italy), after extraction with 2 M KCl for 30 min at 25 °C. To determine SAP, it was extracted with 0.5 M NaHCO_3_ in a 1:5 ratio (*w*/*v*) and analyzed by a UV–visible spectrophotometer (UV-2550; SHIMADZU Corp., Kyoto, Japan). To determine soil pH, 5 g of air-dried soil was used in a slurry ratio of 1:2.5 (soil:CO_2_-free deionized water; *w*/*v*). For TSN, it was determined using 0.5-g air-dried soils, with a 5-mL concentrated sulfuric acid oxidation elimination carried out at 370 °C for 4 h using the SmartChem^®^ 140 Discrete Analyzer. The TSP was determined to use 0.1-g air-dried soil, successively adding 5-mL of concentrated sulfuric acid and 10 drops of perchloric acid three times at 370 °C. The clarified solution was analyzed using a SmartChem^®^ 140 Discrete Analyzer. TSC was determined using 20 mg of air-dried soil by fast combustion at high temperature, using a TOC elemental analyzer (Elementar Analysensysteme GmbH, Langenselbold, Germany). Soil water content was determined gravimetrically, with 10-g of fresh soil placed in an aluminum dish and oven-dried for 48 h at 105 ± 1 °C.

### 2.5. Measurement of AM Fungal Biomass

Soil AM fungal biomass was quantified using phospholipid fatty acid (PLFA) analysis, which is an efficient way to detect soil microbial biomass across different kingdoms, based on the methodology of Bligh and Dyer (1959), as described by Bardgett et al. (1996) [57,58]. Following Bossio and Scow (1998), PLFA concentrations were determined using 8-g of frozen soil to extract the lipids accordingly [59]. Fatty acid methyl esters were separated, quantified, and identified using capillary gas chromatography, with the PLFA analysis conducted on an Agilent 6890 gas chromatograph (Agilent Technologies, Santa Clara, CA, USA). The individual FAs were identified according to the MIDI Sherlock Microbial Identification System (MIDI Inc., Oakland, NJ, USA), and each was quantified using FAME 19:0 (Matreya Inc., State College, PA, USA) as an internal standard. The obtained 16:1ω5c FA concentrations served as indicators of AM fungal biomass [60].

### 2.6. Soil DNA Extraction, PCR, and MiSeq Sequencing

Soil DNA was extracted from frozen soils (each 0.5 g) with the Magnetic Soil and Stool DNA Kit (Tiangen Biotech, Beijing, China), according to the manufacturer’s instructions. The fungi-specific primer sets AML1/AML2 and AMV4.5NF/AMDGR were used to amplify the AM fungal sequences [61,62]. The PCR amplification and library preparation are described in detail in the Appendix A Section. The ensuing amplicons were sequenced (2 × 300-bp paired-end reads) on an Illumina MiSeq sequencer (Shanghai Biozeron Biological Technology Co., Ltd., Shanghai, China).

### 2.7. Bioinformatics

After sequencing, the raw sequences were demultiplexed and quality-filtered using the FASTP pre-processing tool [38]. Next, FLASH software (version 1.2.11) was used to merge the reads with the criteria described by Zheng et al. (2018) [63], after which any singleton OTUs and chimeras were removed [64]. The operational taxonomic units (OTUs) were clustered at 97% similarity levels by UPARSE [65]. A representative sequence of each OTU was then selected and blasted against the AM fungal-specific MaarjAM online database [66,67]. Those sequences are specific to AM fungi and were manually picked up. Finally, these selected sequences were rarefied to the depth of the smallest sample using the ‘rarefy’ function, with a step-size of 20 iterations (Appendix A). The AM fungal community diversity was expressed at the phylotype level using the Shannon–Weiner index. For AM fungal community structure, in-depth analyses were performed by selecting phylotypes with >0.1% sequences per sample [68].

### 2.8. Statistical Analyses

All statistical analyses described below were performed using the R computing platform (R v4.02; R Core Team 2020, https://www.R-project.org/, accessed on 1 April 2024). All data were log-transformed to meet parametric assumptions of normality and homogeneity of variance. Firstly, the plant traits SLA, LNC, LPC, N:P, and Pn were used to calculate functional diversity (i.e., FRic, FEve, and FDis) and CWM by implementing the dbFD function in the ‘FD’ package for R [69]. Significant differences among the three alpine meadow sites in the response variables, including soil properties, functional diversity, community-weighed trait means, AM fungal diversity, AM fungal biomass, and the relative abundance of the top-four dominant genera, were determined by univariate one-way ANOVAs followed by Tukey’s HSD test (at an alpha level of 0.05). To examine how the AM fungal community responded to SWC, we used the Shannon–Wiener index and PLFA markers to assess its diversity and biomass, respectively [63]. To predict the variation in AM fungal diversity and biomass, those variables that differed significantly across sites were chosen for a correlation analysis. Among these, the variables significantly correlated with AM fungal diversity or biomass were then submitted to model selection using the R package ‘MuMIn’ (Version 1.47.1) [70], whose dredge function automatically constructs a complete model set with all possible combinations. All candidate models were compared using the corrected Akaike Information Criterion (AICc), which is adjusted for small sample sizes; the model with the lowest AICc was deemed the best-fitting model [71]. Its determination coefficient (R^2^) was obtained using the r.squaredLR function in the ‘MuMIn’ package. To examine the effect of the alpine meadow site on AM fungal community structure, significant differences among the three sites were examined using the adonis function, with 999 permutations (in the ‘vegan’ package for R, Version 1.47.1; likewise for the other functions). AM fungal structure was evaluated by in-depth analysis at the phylotype level, based on a Bray–Curtis distance matrix using the relative abundance (>0.1%) of each phylotype sequence per sample. Additionally, variables significantly affected by site were used to predict AM fungal community structure through a partial distance-based redundancy analysis (db-RDA), implemented using the capscale function. The best-fitting model was built using the ordistep function (999 permutations), and its significance tested using the anova function (999 permutations).

## 3. Results

### 3.1. Responses of Soil Properties Across the Three Alpine Meadow Sites

All seven soil properties differed among the three sites (Table 1), with the pattern for SWC (F(2, 27) = 152.4, *p* < 0.0001) similar to their precipitation disparities (i.e., HY > MQ > HZ). In particular, TSC and TSN were markedly higher in HY than in MQ and HZ, while STP, SAN, and SAP were evidently higher in HY than in HZ. In contrast, soil pH was lower (acidic) in HY and MQ than in HZ (Table 1).

### 3.2. Responses of the AM Fungal Community Across the Three Sites

Both HY and HZ sites were co-dominated by *Glomus* and *Acaulospora*, yet only *Glomus* was predominant in HZ (Figure 1a). We further examined the responses of the dominant *Glomus*, *Acaulospora*, *Scutellospora,* and *Paraglomus* genera; except for *Scutellospora*, they all exhibited significant differences among the three sites (Figure 1b–e). Specifically, a lower SWC (soil water content) enhanced the relative abundance of *Glomus* significantly (Figure 1b), while that of *Acaulospora* as well as *Paraglomus* rose when going from HY to MQ and HZ (Figure 1b,e). Evidently, the AM fungal diversity changed differently among sites, with median values being about two times higher in HY than in HZ (Figure 2a), and so did AM fungal biomass (Figure 2b), albeit to a lesser extent, with it marginally lower in MQ than in HZ. Furthermore, as seen in Figure 3, the AM fungal community structure differed significantly among the three sites, for which the first and second axes in the db-RDA explaining 65.36% and 17.19% of the total variation, respectively.

### 3.3. Plant Community Functional Diversity and Community-Weighed Trait Means

Soil significantly altered plant community functional diversity except for FRic (Figure 4a). Plant community functional diversity, when expressed as either FEve or FDis, gradually decreased from HY to MQ and HZ (Figure 4b,c), whereas FRic was not significantly different among sites (*p* > 0.1; Figure 4a). We next examined the community-weighed trait means, finding all of them, except for LNC, significantly affected by site, (Figure 5a–e). Notably, in going from HY to HZ, the SLA increased by about 20% (Figure 4a), while Pn decreased by about 15% (Figure 5b). In contrast, LPC was about 33% higher in MQ than the HY and HZ sites (Figure 5d), leading to a marked decrease in the N:P ratio at MQ (Figure 5e).

### 3.4. The AM Fungal Community in Relation to Plant Community Functional Diversity, Community-Weighed Trait Means, and Soil Properties

We used correlations, model selection, and db-RDA to predict the AM fungal community response across the three sites. Evidently, most soil properties were significantly correlated with AM fungal diversity, while the plant community-weighed trait means for LPC and N:P were both significantly correlated with AM fungal biomass (Figure 6). Model selection demonstrated that SAP and STN together explained 47% of the variation in AM fungal diversity, while LPC alone explained 15% of the variation in AM fungal biomass (Table 2). Moreover, the db-RDA in Figure 3 revealed that TSC, N: P, Pn, and FDis collectively explained 40.25% of the variation in AM fungal community structure (F_(4, 25)_ = 4.21, *p* = 0.001).

## 4. Discussion

We assessed key aspects of the AM fungal community, namely its diversity, biomass, community composition, and structure, in the alpine meadow ecosystem of the Tibetan Plateau. Our results revealed remarkable changes in AM fungal diversity, plant community functional diversity, and soil nutrients across three alpine meadow sites. The higher precipitation at HY led to a higher SWC, which was associated with a substantial increase in its AM fungal diversity. This result agrees well with the work by Zhang et al. (2016) [72], who found that AM fungal diversity gradually fell as precipitation declined in the alpine steppe of the Tibetan Plateau. A recent study showed that long-term drought in an alpine steppe markedly reduced its AM fungal diversity [73], whereas a similar disturbance had negligible effects in mesic grassland [33]. Conversely, a 7-year period of increased precipitation led to greater AM fungal richness in the steppe grassland of Inner Mongolia [31]. These results indicate that how AM fungal diversity responds to changes in precipitation might depend on a grassland ecosystem’s properties, especially its ambient soil nutrients. Here, our correlations and model selection demonstrated that SAP and TSN could positively predict the AM fungal diversity of alpine meadows. Our results are thus consistent with those of Liu et al. (2012) [74], who found that a combination of nitrogen (N) and phosphorus (P) added at a low dose enhanced AM fungal diversity as well as root colonization in an alpine meadow, but reduced both features when these nutrients were applied at a high dose. Alongside our results, this suggests that AM fungi are N- and P-limited in alpine meadows.

This dual limitation should spur AM fungal taxa to compete for available N and P in soils because AM fungal tissue requires more N and P than do typical plant tissues. This intensified competition could lead to a stark reduction in AM fungal diversity. However, since the threshold for nutrient limitation tends to be lower for AM fungi than for plants [75], low-dose N and P additions often ease AM fungal competition for N and P, leading to an increase in AM fungal diversity [76,77]. By contrast, high-dose N and P additions can offset the N and P limitations faced by various plant species [78]. Plant hosts will reduce their C allocation to AM fungi, which would allow AM fungal taxa to compete for derived-plant C, leading to less AM fungal diversity [79,80]. We find that both AM fungal diversity and biomass differed across three alpine meadow sites. Notably, AM fungal biomass is slightly higher in MQ than in HZ. This could be due to between-site discrepancies in plant community functional composition, especially the relative abundance of grasses and forbs [81]. In MQ, the high accumulative relative abundance of forb species may spur the plant community to demand more P, because the LPC is markedly higher in forbs than grasses, and forbs with thick roots depend more on AM fungal hyphal networks for nutrient capture than do grasses whose roots are thin [82,83]. Hence, the AM fungal biomass predicted by the plant community weighted-LPC in our study could be explained by the mean of cumulative relative abundance of forbs exceeding 52%, along with their higher LPC values (Appendix A, Figure 5d). We recorded the presence of 36 forb species, constituting 73.47% of all 49 plant species observed in the three surveyed alpine meadow sites. These forbs mostly rely on the C_3_ photosynthesis pathway, and more than half of them are native perennial plants typical of alpine meadow communities on the Tibetan Plateau (Appendix A).

We also found that fungal members of *Glomus* dominated the HZ site. According to work by Egerton-Warburton et al. (2007) [76], *Glomus* is predominant in semiarid sites, while *Scutellospora*, *Gigaspora*, and *Acaulospora* are abundant in mesic sites of grasslands. Recently, we observed that long-term drought bolstered the relative abundance of *Glomus* in an alpine steppe [73]. This varied response of AM fungal taxa or clades to soil moisture levels may partly arise from their life history strategies based on functional traits [84]. It is known that certain AM fungal taxa or clades differ in their C demands, C-use efficiency, and hyphae-related traits [85]. For example, members of *Glomus* are more effective at C utilization; so, despite less available C, they could maintain their hyphal growth and sporulation [86].

Long-term drought often depresses photosynthetic activity in the canopy of plants and reduces their C allocation to belowground to host AM fungi [87], possibly favoring the predominance of *Glomus*. The C demands of distinct AM fungal taxa are also thought to be trait-linked [84]. For example, those AM fungal taxa with delicate hyphae and small spores require less C [85]. Although a trait-based framework has been theoretically proposed to explain the responses of AM fungal clades to environmental changes [88,89], empirically tracking in situ the C allocated to differing AM fungal taxa and evaluating individual AM fungal traits are both daunting tasks. At this point, we can only surmise that, with climatic changes, AM fungal communities maximize their own fitness through coordinated shifts in their composition [90].

Another important finding of our study was that variation in AM fungal community structure is predicted well by plant community functional composition as well as soil properties. Several studies have shown that changed precipitation regimes can alter AM fungal community structure, but not AM fungal diversity [30,33]. This would suggest a pivotal role of AM fungal community structure that is site-responsive, and the changes in AM fungal community structure may have important consequences for ecosystems long before AM fungal diversity is threatened by extinction. Accordingly, it is crucial to adequately consider AM fungal community structure for grassland management, conservation, and restoration in the face of ongoing climate change.

We find that plant traits involved in the C-nutrient economy can predict the variation in AM fungal community structure. The best predictors were the plant community-weighed N:P ratio and Pn, along with the TSC. In previous models, only the effects of soil N:P ratio on AM fungal communities were predicted [79,80]. Leaf N:P ratios may be used to gauge the degree of plant nutrient availability and limitation, and they play an essential role in photosynthesis processes [91,92]. Few studies, however, have specifically focused on how the leaf N:P ratio at the plant community level may affect the AM fungal community [78]. Due to their ramified hyphae, AM fungi can transfer P outside the depletion zone and some AM fungal taxa can directly uptake organic P [93]. In contrast, AM fungal hyphae may not provide an N-uptake advantage over roots due to the greater mobility of nitrate and ammonium in soils [79]. Yet a recent meta-analysis did uncover a positive effect of AM fungi on rates of N uptake [20]. Several lines of evidence have clearly demonstrated that AM fungal C availability is able to stimulate both N and P uptake and transport in the hyphal network [27,94]. Therefore, plant traits critically involved in nutrient-C exchange via AM fungal symbionts may figure prominently in the community assembly of AM fungi.

As a major contributor to soil organic C, photosynthesis directly provides C for AM fungal growth and development [29]. Indeed, the AM fungi-derived C from plants constitutes, on its own, an important fraction of TSC [37]. Moreover, AM fungal hyphae also exude various C compounds into soils, such as sugars, carboxylates, and amino acids [72]. We also observed that FDis is a significant predictor of AM fungal community structure in the best-fitting model. The FDis reflects the degree of trait dispersion within a plant community, whereby a high FDis augments overall resource utilization (e.g., of water, light, and nutrients) via trait differentiation [12], and this conceivably may further affect the community structure of AM fungi. The number of these plant species associated with soil nutrients declined sequentially from HY to MQ, and finally to HZ (Appendix A; Table 1). Hence, our results provide compelling evidence that plant functional composition and soil properties act together to shape the AM fungal community of alpine meadows on the Tibetan Plateau.

Although our study demonstrates clear compositional shifts in AM fungal communities along environmental gradients, we did not collect functional trait data of vascular plants that would enable a direct assessment of climate-driven assemblage shifts. However, previous studies suggest that changes in temperature and precipitation regimes can lead to parallel restructuring of both aboveground and belowground communities [95]. The observed elevation-associated turnover in both vascular plants and AM fungal taxa may thus reflect coordinated responses to environmental filtering, although more integrative, long-term data would be required to confirm this.

Given the obligate symbiotic relationship between AM fungi and vascular plants, projected shifts in plant species richness under climate change are likely to cascade into changes in AM fungal diversity and community structure. While long-term monitoring data are lacking, studies suggest that declines in dominant mycotrophic plants or altered plant community compositions due to climate shifts could reduce the availability of compatible hosts, thereby constraining AM fungal diversity and function. Future climate scenarios should therefore consider plant–fungal interactions to more accurately forecast biodiversity trajectories in alpine grasslands.

## 5. Conclusions

Our study shows that the diversity of AM fungi, as well as their community composition and structure, can vary markedly across alpine meadow sites, while their biomass tends to remain stable across space. The high soil water content induced by more precipitation is capable of maintaining greater plant functional diversity, leading to changed soil nutrients and AM fungal diversity, whereas *Glomus* prevails in site conditions of low soil water content and low soil nutrient availability. Potential differences in traits of AM fungi or their life history strategies could impact the responses of AM fungal community composition in alpine meadows. Our study highlights the importance of considering both plant traits and soil properties in tandem, particularly those most closely associated with nutrient–C exchange via AM fungal symbionts, in shaping AM fungal communities in the alpine meadow ecosystem of the Tibetan Plateau. Our findings have valuable implications for further understanding the response of AM fungi and their diversity to nutrient–C exchange dynamics tied to plant traits and soil properties in grasslands.

## Figures and Tables

**Figure 1 jof-11-00337-f001:**
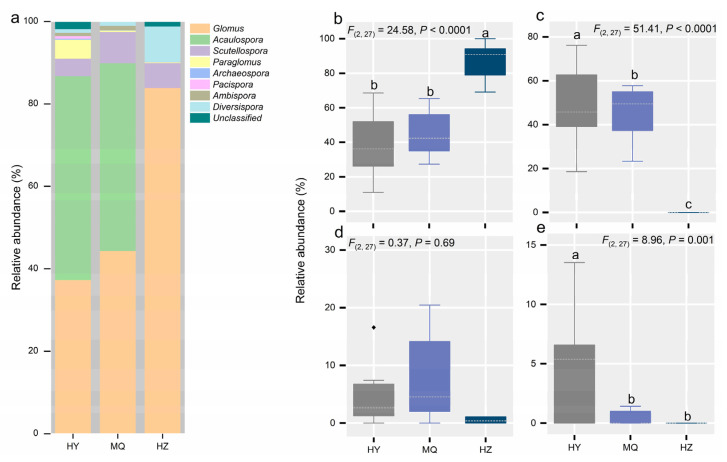
Changes in AM fungal community composition (**a**), and dominant genera *Glomus* (**b**), *Acaulospora* (**c**), *Scutellospora* (**d**), and *Paraglomus* (**e**) across three alpine meadow sites on the Tibetan Plateau. The boxplots show the median (dashed white line) and quartiles; in (**d**), the black diamond is an outlier value. Different letters indicate significant differences between sites. We labeled significant differences (*p* < 0.05) with lowercase letters, otherwise, no labels are shown. Hongyuan (HY), Maqu (MQ), Hezuo (HZ).

**Figure 2 jof-11-00337-f002:**
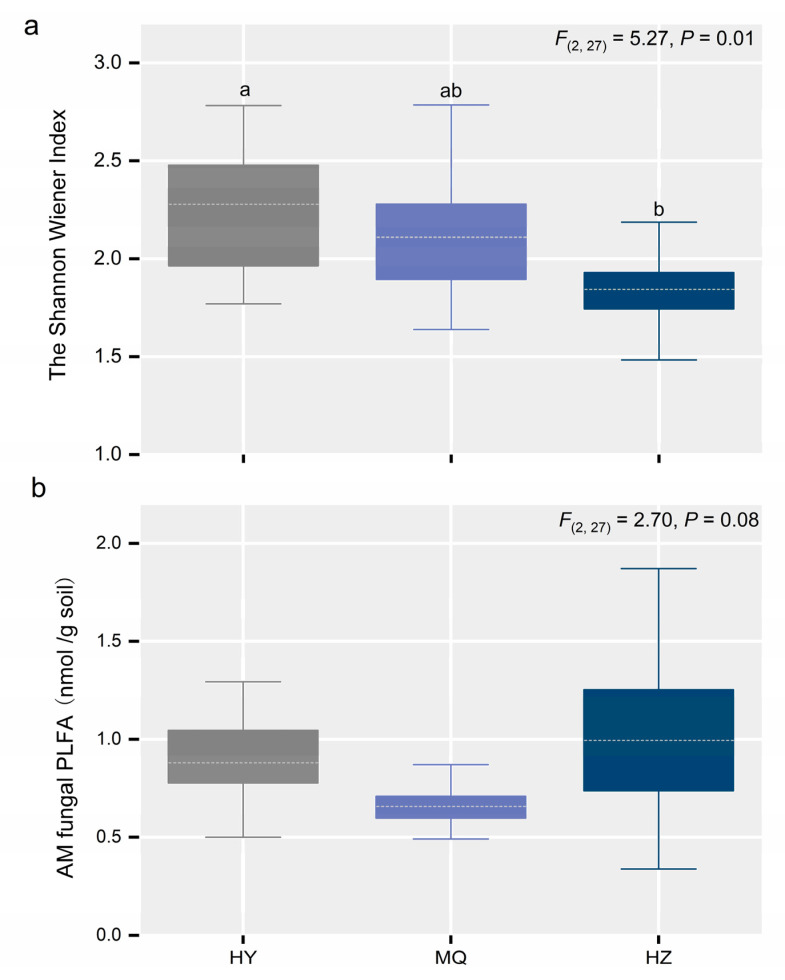
Changes in the AM fungal diversity (**a**) and biomass (**b**) across the three alpine meadow sites. Different letters indicate significant differences between them. The boxplots show the median (dashed white line) and quartiles. We labeled significant differences (*p* < 0.05) with lowercase letters, otherwise, no labels are shown. Hongyuan (HY), Maqu (MQ), Hezuo (HZ).

**Figure 3 jof-11-00337-f003:**
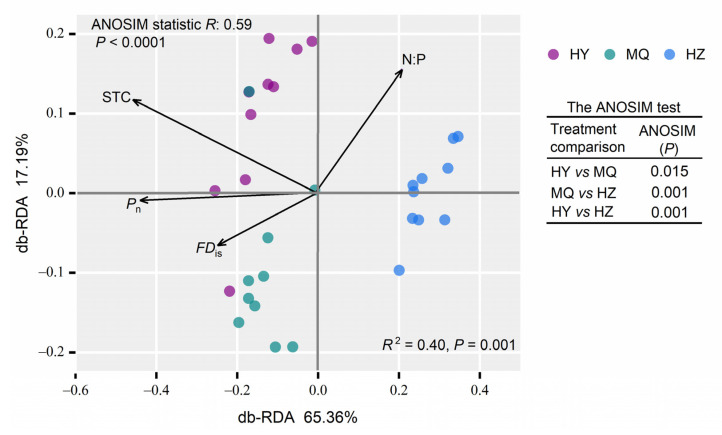
A distance-based redundancy (db-RDA) analysis of AM fungal community structure across three alpine meadow sites (HY, MQ, HZ). Total soil carbon (TSC), the plant community-weighed-N:P ratio mean (N:P), community-weighed photosynthetic rate mean (P_n_), and plant community functional dispersion (FD_is_) significantly predicted AM fungal communities, based on a stepwise model selection using permutation testing. Hongyuan (HY), Maqu (MQ), Hezuo (HZ).

**Figure 4 jof-11-00337-f004:**
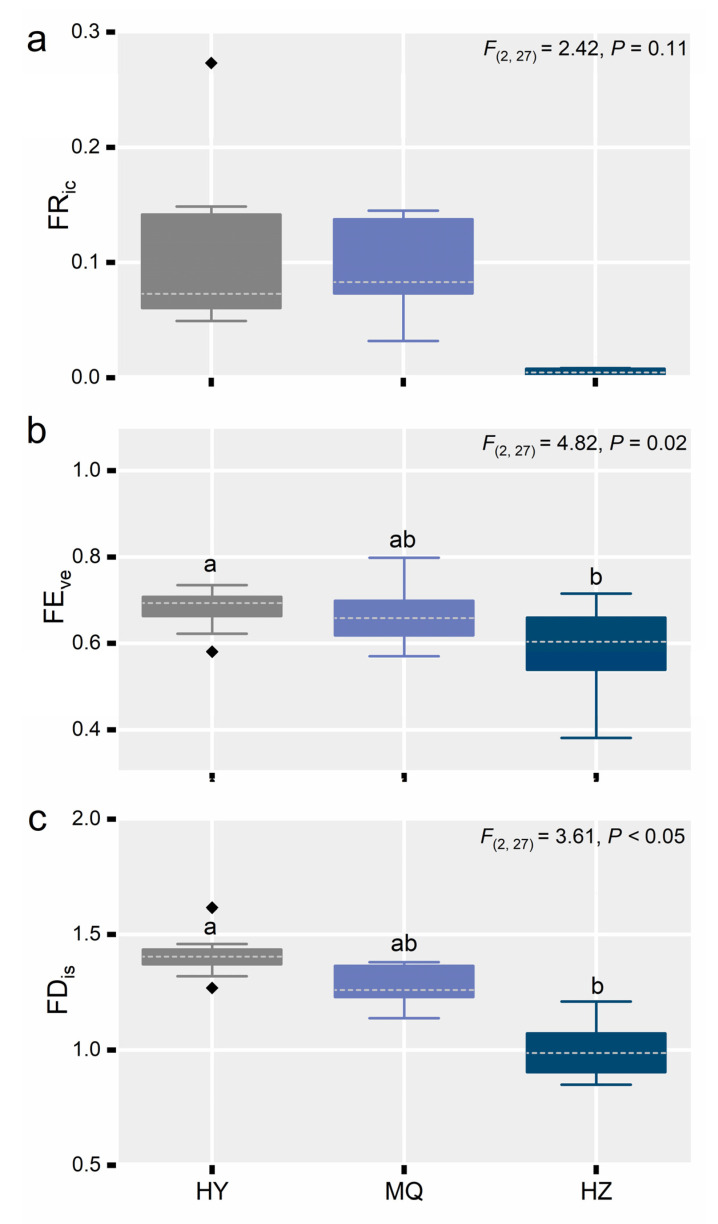
Changes in plant community functional diversity across the three alpine meadow sites, in terms of functional richness (FR_ic_) (**a**), functional evenness (FE_ve_) (**b**), and functional dispersion (FD_is_) (**c**). Different letters indicate significant differences between sites. The boxplots show the median (dashed white line) and quartiles; black diamonds are outlier values. We labeled significant differences (*p* < 0.05) with lowercase letters, otherwise, no labels are shown. Hongyuan (HY), Maqu (MQ), Hezuo (HZ).

**Figure 5 jof-11-00337-f005:**
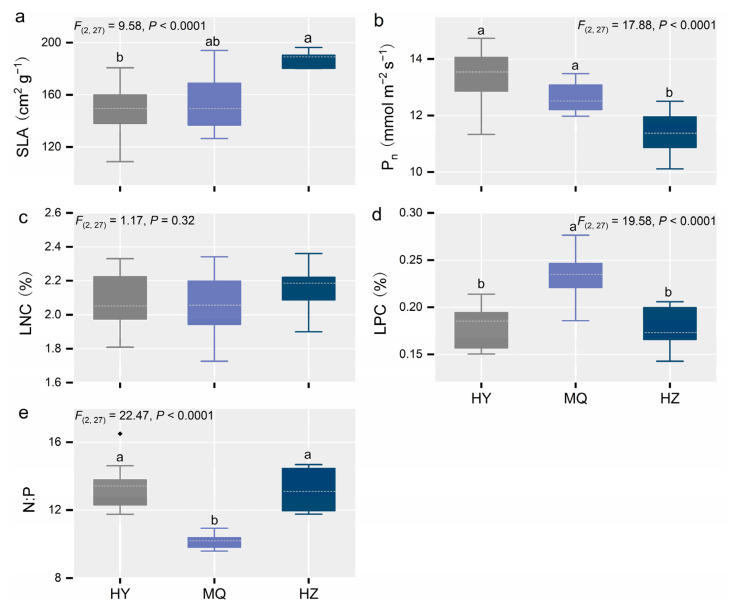
Changes in plant community-weighed trait means across the three alpine meadow sites, for the specific leaf area (SLA) (**a**), photosynthetic rate (P_n_) (**b**), leaf nitrogen concentration (LNC) (**c**), leaf phosphorus concentration (LPC) (**d**), and nitrogen-to-phosphorus concentration ratio (N:P) (**e**). Different letters indicate significant differences between sites. The boxplots show the median (dashed white line) and quartiles; the black diamond is an outlier value (**e**). We labeled significant differences (*p* < 0.05) with lowercase letters, otherwise, no labels are shown. Hongyuan (HY), Maqu (MQ), Hezuo (HZ).

**Figure 6 jof-11-00337-f006:**
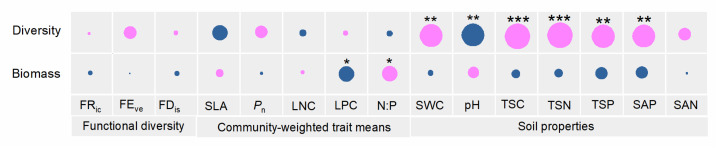
Pearson correlations between AM fungal diversity and biomass with plant community functional diversity, community-weighted trait means, and soil properties. Pink indicates positively correlated and blue indicates negatively correlated. A circle’s size corresponds to the magnitude (*r* value) of that correlation. *, *p* < 0.05; **, *p* < 0.01; ***, *p* < 0.001. FR_ic_—functional richness; FE_ve_—functional evenness; FD_is_—functional dispersion; SLA—specific leaf area; LNC—leaf nitrogen concentration; LPC—leaf phosphorus concentration; N:P—leaf nitrogen and phosphorus concentration ratio; P_n_—photosynthetic rate; SWC—soil water content; TSC—total soil carbon; TSN—total soil nitrogen; TSP—total soil phosphorus; SAP—soil available phosphorus; SAN—soil available nitrogen.

**Table 1 jof-11-00337-t001:** The results of one-way ANOVAs for soil properties among three sites.

Soil Properties	Sites		
	HY	MQ	HZ
Soil water content (%)	55.47 ± 1.70 a	35.65 ± 0.99 b	28.57 ± 0.48 c
Soil pH	5.97 ± 0.10 b	6.34 ± 0.13 b	6.93 ± 0.09 a
Total soil carbon (g Kg^−1^)	64.06 ± 1.54 a	53.14 ± 2.76 b	35.78 ± 0.77 c
Total soil nitrogen (g Kg^−1^)	5.74 ± 0.13 a	4.96 ± 0.26 b	3.47 ± 0.07 c
Total soil phosphorus (g Kg^−1^)	0.84 ± 0.08 a	0.78 ± 0.07 ab	0.56 ± 0.05 b
Soil available phosphorus (mg Kg^−1^)	3.6 ± 0.36 a	3.33 ± 0.27 ab	2.42 ± 0.20 b
Soil available nitrogen (mg Kg^−1^)	4.59 ± 0.37 a	4.05 ± 0.39 ab	3.22 ± 0.35 b

Note: Different letters indicate significant differences between sites. Values shown are mean ± SE, n = 10. Hongyuan (HY), Maqu (MQ), Hezuo (HZ).

**Table 2 jof-11-00337-t002:** Best-fitting models for AM fungal diversity and biomass, selected using the ‘dredge’ function.

Model	Selected Variables	*df*	*R* ^2^	ALCc	ΔAICc	Weight (*wi*)
Diversity	Soil available P, Soil total N	4	0.47	−106.4	0.00	0.08
Biomass	Leaf P concentration	3	0.15	−66.6	0.00	0.42

Note: P (phosphorus), N (nitrogen), ALCc (Akaike Information Criterion corrected), ΔAICc (delta AICc), Weight (*wi*) (Akaike weight).

## Data Availability

The original contributions presented in this study are included in the article/Appendix A. Further inquiries can be directed to the corresponding author.

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
