# Peer review of "Plant–Soil Interactions Shape Arbuscular Mycorrhizal Fungal Diversity and Functionality in Eastern Tibetan Meadows"

_jof, 2025, doi:10.3390/jof11050337_

Round 1
Reviewer 1 Report
The revised manuscript is solid in content, both in its approach and analysis. It shows strong evidence of the plant-soil relationship as a control of the AM fungal community. It uses molecular tools to characterize AM fungi. The information is relevant as a reference framework for future studies worldwide, especially in Eastern Tibetan. The issue is consistent with the Journal. A few aspects were identified throughout the manuscript that need to be improved or clarified. This is a well-supported study, so in the discussion, I only offer a few ideas for the authors' consideration. The aspect regarding AM fungi and climate change and precipitation could be better supported with its data by looking for a correlation of the mycorrhizal fungal community with precipitation levels, or by reporting whether the correlations recorded between variables change between sites with different precipitation levels and soil moisture. My recommendation is that the manuscript should be published once the authors resolve (or argue) all the aspects pointed out throughout the manuscript.
The revised manuscript is solid in content, both in its approach and analysis. It shows strong evidence of the plant-soil relationship as a control of the AM fungal community. It uses molecular tools to characterize AM fungi. The information is relevant as a reference framework for future studies worldwide, especially in Eastern Tibetan. The issue is consistent with the Journal. A few aspects were identified throughout the manuscript that need to be improved or clarified. This is a well-supported study, so in the discussion, I only offer a few ideas for the authors' consideration. The aspect regarding AM fungi and climate change and precipitation could be better supported with its data by looking for a correlation of the mycorrhizal fungal community with precipitation levels, or by reporting whether the correlations recorded between variables change between sites with different precipitation levels and soil moisture.

Author Response
Thank you very much for taking the time to review our manuscript. Please see the detailed response below, and we have highlighted the changes in red in our resubmission of the manuscript for your review.
Reviewer: 1
REVIEW REPORT FORM
Some results are presented in the Materials and Methods section (Table 1) and should therefore be relocated. You should also take care with the spelling of scientific names. Other simple aspects are also brought to the authors' attention.
Response: We greatly appreciate the positive comments on our manuscript. We have moved Table 1 to the results of revised manuscript (Lines 310 – 312).
The English could be improved to more clearly express the research.
Response: Thank you for reviewer’s valuable comment. We have carefully revised the manuscript to improve the clarity and fluency of the English language throughout the manuscript. The revised version has been proofread by a native English speaker. All changes are highlighted in the revised manuscript in read colors.
COMMENTS FOR AUTHORS
The revised manuscript is solid in content, both in its approach and analysis. It shows strong evidence of the plant-soil relationship as a control of the AM fungal community. It uses molecular tools to characterize AM fungi. The information is relevant as a reference framework for future studies worldwide, especially in Eastern Tibetan. The issue is consistent with the Journal. A few aspects were identified throughout the manuscript that need to be improved or clarified. This is a well-supported study, so in the discussion, I only offer a few ideas for the authors' consideration.
Response: We greatly appreciate the positive comments and valuable suggestions on our manuscript.
The aspect regarding AM fungi and climate change and precipitation could be better supported with its data by looking for a correlation of the mycorrhizal fungal community with precipitation levels, or by reporting whether the correlations recorded between variables change between sites with different precipitation levels and soil moisture. My recommendation is that the manuscript should be published once the authors resolve (or argue) all the aspects pointed out throughout the manuscript.
Response: Thank you for this insightful comment. We have discussed of the potential influence of soil moisture on AM fungal community composition and diversity (Figure 3, Table 2). While our current sampling design does not allow for detailed statistical correlation between precipitation and fungal diversity due to the limited number of sites. So only 10 soil water content data obtained from each sample plot could be used for result analyzing. We have analyzed best-fitting models selected using the function ‘dredge’ for AM fungal diversity and biomass with soil variables. We did not selected the soil water content significant correlation with Am fungal diversity and biomass. we acknowledge the importance of precipitation as a driver of AM fungal assembly.
Finally, we thank you for your constructive comments and valuable suggestions on our manuscript. We believe that the quality of the revised manuscript has improved markedly, and hope the revised manuscript is acceptable for publication in Journal of Fungi.
Regards,
All co-authors
Reviewer 2 Report
Plant-Soil Interactions Shape Arbuscular Mycorrhizal Fungal
Diversity and Functionality in Eastern Tibetan Meadows
The references included and cited are relevant and represent a good review of some areas of the topic and sites sampled. It would be helpful to have a broader discussion in the Introduction that would set the stage for the good study that was undertaken. There should be a stronger presentation of the description of the vascular plant community and the reflective mycosphere the paper examines. It should present the numbers of vascular plant species recorded at each of the three areas – with links to plant species lists for the sites, and, also, the accompanying number of AM mycorrhizal taxa (and what are they?) and with which vascular plant species is each AM taxon associated. Without some numbers it is difficult to see what the authors mean when they talk about mycorrhizal diversity. Descriptions of climate-change related assemblages in the AM mycorrhizae should be reflected in the vascular plant community. I think that before the paper can be accepted the revision must address these issues and others mentioned below. Likely there are published papers that can be simply cited that already present most of this (species lists, known symbiotic relationships, and so forth). They could be referenced without reproducing all the already published data. The experiments and their implementation are well-designed and implemented – and are a strong, original contribution to our knowledge of this subject in Tibet. The conclusions are supported by the data presented and the paper will be suitable for publication upon revision.
The use of the English language is excellent, and I highlighted in red some suggested wording changes for consideration in the narrative.
Can the future species richness of the vascular plant symbionts/hosts be predicted with increasing impacts from climate change – and wouldn’t that deeply influence the fungal symbiont as well? Since they are symbiotically linked to each other, wouldn’t the vascular plant host be important in determining future and reflective fungal diversity and presence? As mentioned above, it would be useful to have links to vascular plant species checklists for the sites, and an indication of which are known to be symbiotic (and with which AM taxa). If there are particularly important vascular plant species for AM taxa, their likely future abundance or decrease should be mentioned.
Some potentially useful resources that might be included are:
Index Fungorum https://www.indexfungorum.org/ and the CABI databases https://www.speciesfungorum.org/ and Species Fungorum https://www.speciesfungorum.org/Names/Gsd.htm
Some of the herbaria that have strong collections of vascular plants from Tibet include the following (no doubt there are others, as well), and if vouchers were collected of any plants, they could be placed in one or more these herbaria:
Kunming Institute to Botany (KUB) https://sweetgum.nybg.org/science/ih/herbarium-details/?irn=125083
Arnold Arboretum (A) https://arboretum.harvard.edu/research/herbarium-of-the-arnold-arboretum/#:~:text=Arnold%20Arboretum%20(A)-,Herbarium%20of%20Cultivated%20Plants,grounds%2C%20are%20also%20housed%20here.&text=Management%20and%20development%20of%20the,Herbarium%20Collections%20Policy%20%5Bpdf%5D.
British Museum (Natural History) (BM) https://sweetgum.nybg.org/science/ih/herbarium-details/?irn=126148#:~:text=Name,hunnex@nhm.ac.uk
Sino-Himalayan Plant Association
Kew Royal Botanic Gardens. Plants of the World .https://powo.science.kew.org/taxon/urn:lsid:ipni.org:names:26916-2#higher-classification
There are likely others, as well – but placing representative vouchers or duplicates of collections would make it more facile for other researchers to access – and would expand the scientific value of the paper. Presumably, deposited voucher specimens would be phenologically scored, digitized, and georeferenced by the receiving herbarium.
In addition to vascular plant host plants, did the researchers collect and place any reference voucher specimens of the AM mycorrhizal species in a fungorium or an herbarium (or other repository site) so that they could document their data and others could examine them? Are there collections they could cite and annotate that are representative the species they studied from the area of their sample sites?
If voucher AM collections were made during this study, it might make them more available (virtually) through sites such as Fungi: Mycology Collections Portal (MyCoPortal) and https://www.mycoportal.org/portal/ - for North America. An analog site that would include Tibet could be used
A link that might be useful to the researchers is Mycocsom - JGI Mycorrhizal Genomics Initiative (MGI) project.
JGI MycoCosm https://mycocosm.jgi.doe.gov/Mycorrhizal_fungi/Mycorrhizal_fungi.info.html#:~:text=Mycorrhizal%20Fungi&text=Within%20the%20framework%20of%20the,MGI%20master%20paper(s).&text=Please%20confirm%20that%20you%20want,changes%20for%20'Mycorrhizal%20Fungi'.
Another resource is the Fungal Genomics Resource https://jgi.doe.gov/ . It might be possible to place voucher data material through these one or several of these esource bases.
Some specific suggestions for wording changes are indicated in red below. These include italicizing genera and specific epithets, putting a space between reference brackets and the word preceding them, and so forth. The References need to be carefully edited, making sure that in the re-submission, all papers and books are consistently cited and follow the requirements of the journal. I think after addressing the issues mentioned above and the indications in the narrative that the paper will be acceptable for publication, and I encourage the researchers to re-submit it.
Plant-Soil Interactions Shape Arbuscular Mycorrhizal Fungal
Diversity and Functionality in Eastern Tibetan Meadows
Shihu Zhang 1,*, Zhengying Yang 1, Xuechun Yang 1, Xiaoyu Ma 1, Qun Ma 1, Miaojun Ma 2 and Jiajia Zhang 3
1 College of Life Science, Northwest Normal University, Lanzhou Gansu 730070, China; zhang-
shh05@163.com
2 College of Ecology, Lanzhou University, Lanzhou Gansu 730000, China
3 School of Life Sciences, Institute of Life Science and Green Development, Hebei University, Baoding, Hebei
071002, China 9
* Correspondence: zhangshh05@163.com
Abstract: Arbuscular mycorrhizal (AM) fungi occur in the interface between soils and plants. Yet the impacts of the plant community functional composition and soil properties on AM fungal communities remain poorly understood in the face of ongoing climate change. Here, we investigated the AM fungal community in alpine meadow habitats of the Tibetan Plateau by linking fungal species richness to plant community functional composition and soil parameters at three latitudinal sites. We found that AM fungal diversity and plant functional diversity, as well as the contents of soil nutrients, were significantly higher in the southernmost site, Hongyuan (HY). Total soil nitrogen and soil-available phosphorus explained the variation in AM fungal diversity, while AM fungal biomass was best predicted by the plant community-weighed mean nitrogen:phosphorus ratio (CWM-N:P). Glomus species preferentially occurred in the northernmost site of Hezuo (HZ). Distance-based redundancy analysis (db-RDA) revealed that AM fungal community structure was influenced by not only CWM-N:P but also by plant community-weighed mean photosynthetic rate (CWM-Pn), soil total carbon, and plant community functional dispersion (FDis). We conclude that plant traits and soil properties are crucial for nutrient-carbon (C) exchange because fungal symbionts may shape AM communities in this vast alpine meadow ecosystem. Our findings provide a timely insight into AM fungal community assembly from the perspective of nutrient-C exchange dynamics in the Tibetan Plateau’s alpine meadow habitats.
Keywords: arbuscular mycorrhizae; fungal community; plant functional traits; soil properties; Tibetan Plateau
- Introduction
In recent decades, human-induced climate change has contributed to altered precipitation patterns and hydrological cycles [1,2]. In particular, the magnitude, frequency, and duration of heavy precipitation events are being enhanced significantly by greater warming at global, continental, and regional scales [3,4]. Grassland ecosystems are an integral terrestrial component and are especially sensitive to precipitation changes, since they are naturally limited by soil water availability [5]. Nevertheless, reduced rainfall such as that leading to drought, are also altering the composition and structure of grassland plant communities, further affecting ecosystem functioning and resiliency [6,7,8]. Plant community functional composition, whether defined by the single trait (i.e., community-weighed trait means, CWMs) and/or multiple traits approach (i.e., trait distribution within communities), has been linked to the responses of grassland ecosystems to altered precipitation regimes [9,10,11]. High levels of functional trait diversity can often bolster overall resource utilization via complementarity dynamics among species, conferring greater resiliency to plant communities facing environmental changes [12,13]. Moreover, certain plant traits can mediate how grassland communities impact soil mois- ture [14]. For instance, a tall canopy limits the amount of light radiation reaching the ground surface, leading to more soil water retained. Although plant traits are thought to play an essential role in soil functions, the linkages between plant community functional composition and the soil microbial community are far from fully understood [15].
Arbuscular mycorrhizal (AM) fungi (Glomeromycota family), can form mycorrhizal symbioses with the roots of about 72% of terrestrial plant species [16], which constitute at least 55% of terrestrial vegetation’s cover [17] and 63% of its net primary productivity [18]. While AM fungi are widely distributed in all global regions, they are most prevalent in low and mid-latitude forest and grassland ecosystems, where they figure prominently in plants’ uptake of mineral nutrients and soil water provisioning [19,20,21]. Since AM fungi can establish mutualistic associations with most land plants, it links the aboveground and belowground parts of an ecosystem through complex mineral-nutrient-carbon exchanges, and this applies to most grasslands and their species [22,23]. Besides directly affecting plants’ photosynthetic intensity by influencing their carbon’s source-sink relationship, mycorrhizal symbiosis can augment the shift to a belowground carbon sink from plants; this process reduces photosynthate accumulation in plants, inducing them to increase their photosynthetic rate to compensate for the carbon transferred [24,25]. For example, Gavito et al. (2019) found that plant photosynthetic rate was reduced by 10% to 40% within a short period of disconnecting a portion of their AM fungal hyphae [26]. This strongly suggests that mycorrhizal symbiosis can modulate the source-sink relationship of photosynthesized carbon in plants, which in turn drives the intensity of photosynthesis. AM fungi can improve plant fitness through the provision of nutritional and non- nutritional benefits for their hosts [20,27]. Recently, there is emerging evidence showing that AM hyphae can directly provide host plants with water via roots [28]. In return, their plant hosts provide AM fungi with carbohydrates for the formation, maintenance, and function of mycorrhizal structures. In tandem, AM fungi produce extensive hyphal net- works in roots and soil, intimately linking plants and soils [29].
Several studies have demonstrated that changed precipitation could alter the diversity and composition of AM fungal communities [30,31,32,33,34]. For example, long-term drought increased the AM fungal colonization of roots yet decreased the extraradical mycorrhizal hyphae densities in soils [35]. In reanalyzing published data from grasslands in North America [36], Rillig (2004) found a strong positive relationship (R2 = 0.97) between AM fungal hyphae densities and precipitation levels [37]. The responses of AM fungal communities to precipitation changes are known to associated with precipitation-induced changes in plant community composition [35,31]. In addition to that close association, a plant community’s functional composition has been linked to how it responds to changes in precipitation patterns [9,11]. Alpine grasslands of the Tibetan Plateau are very sensitive to precipitation changes [38,39]. Over recent decades, while the mean annual precipitation in this area has increased sharply [40], the significant warming of the Tibetan Plateau is believed to promote the growth of alpine grasslands, it may also lead to an increase in atmospheric aridity, thereby constraining vegetation growth [39]. Most plants in these alpine grasslands are colonized by AM fungi that are generally abundant [41]. Thus, a better understanding of those mutualistic associations is imperative for informing alpine grassland management and conservation in the face of rapid climate change [40], since AM fungi play a key role in plant nutrient availability, the productivity, and biodiversity of vegetation communities [42,43].
In this study, we used plant community-level traits and soil properties associated with carbon (C) and nutrients to investigate AM fungal community responses in three alpine meadow sites given the following considerations. Previous studies have demonstrated that precipitation-driven shifts in plant community functional composition can influence AM fungal communities [35,31]. Moreover, changes in precipitation patterns have been shown to affect plant community traits, which in turn govern community-level functional composition [9,11]. Additionally, plant-derived C is expected to underpin the fitness of AM fungi because they are asexual obligate biotrophs that rely exclusively on C from host plants to complete their life cycle [29]. The ability of AM fungi to improve the water and photosynthetic profiles of plants is well documented for agricultural and horticultural plants [26,44,45]. In contrast, far less is known about the modulation of photosynthesis by AM fungi vis-à-vis the induced C sink strength mechanism in grassland ecosystems.
We measured responses of AM fungal biomass and diversity, various soil properties, plant community functional diversity (functional richness - FRic, functional evenness - FEve, and functional dispersion - FDis), community-weighted trait means (CWM) of specific leaf area (SLA), leaf nitrogen concentration (LNC), leaf phosphorus concentration (LPC), the ratio of leaf nitrogen and phosphorus ratio (N:P), and photosynthetic rate (Pn) to test the following three hypotheses: (1) Higher levels of soil nutrients would augment fungal diversity; (2) AM fungal biomass would respond to plant traits such as photosynthetic rate and plant community functional dispersion; and (3) both plant traits and soil properties involved in nutrient-C exchange would shape AM fungal community structures in the alpine meadow ecosystem of the Tibetan Plateau.
- Materials and Methods
- Site Selection
We selected three alpine meadow sites at the eastern edge of the Tibetan Plateau (Figure A1). The first site (‘HY’) was located at the Alpine Wetland Ecosystem Field National Observation and Science Research Station of Hongyuan County, in Sichuan Province, China (32o48′ N, 102o33′ E; 3508 m a.s.l.), where the mean annual precipitation is 690 mm and the mean annual temperature is 0.9 °C, with a mean vegetation coverage of 93.81% [46]. The second site (‘MQ’) was located at the Gannan Grassland Ecosystem National Observation and Research Station of Maqu County, in Gansu Province, China (33o38′ N, 101o53′ E; 3500 m a.s.l.), having a mean annual precipitation and temperature of 620 mm and 1.2 °C, respectively [47,48]. There, the mean vegetation cover was lower, at 85.23%, due to rodent activity in the site’s plant community. The third site (‘HZ’) was located at the Gannan Grassland Ecosystem National Observation and Research Station of Hezuo as brach station, in Gansu Province, China (34°55′ N, 102°53′ E; 2960 m a.s.l.), with a mean annual precipitation and temperature of 560 mm and 2.0 °C, respectively [49], whose mean vegetation cover was 91.35%.
Their plant communities are dominated by perennial herbaceous species in the families Compositae, Gramineae, Ranunculaceae, Lamiaceae, Leguminosae, and Cyperaceae that can nevertheless attain a high degree of richness: more than 30 spp. can co-occur within a 50-cm × 50-cm quadrat at these three sites [50,51]. These alpine meadow soils all belong to the Cambisol type (FAO taxonomy) [52]. Taken together, the environmental conditions and vegetation types correspond to typical alpine meadow on the eastern Tibetan Plateau. All three sampling sites were all moderately grazed by yaks from October to May, when serving as winter pasture, with yaks moved to higher elevations from May to mid-September (as summer pasture) [53]. Detailed information on the soil physicochemical properties and plant species of the alpine meadow sites can be found in Table 1 and Table A1, respectively.
Table 1. The results of one-way ANOVAs for soil properties among three sites.
2.2. Field Investigation and Soil Sampling
To examine the plant communities, within a 100-m × 200-m area at each site we ran- domly selected 10 plots (each 10 m × 10 m) at least 2 m apart. For each plant species, its cover within a given plot was visually quantified in mid-August 2020 in a 1-m × 1-m quadrat consisting of a grid formed by 100 equally distributed cells (each 10 cm × 10 cm; whose intersection points were thus 10 cm apart). Community cover was summed across the species present in quadrat, on a per quadrat basis, for which relative species cover was calculated simply as the cover of an individual species divided by the community cover. Plant traits were also surveyed, but only for the most common species in the quadrat (i.e., each having an accumulated abundance > 90%). After plant community sampling, five soil cores (topsoil layer: 0-20 cm depth) were randomly collected from each plot. These soil samples were homogenized, to yield a single composite sample per plot, and then sieved through 2-mm mesh after removing any plant roots and stones. To avoid contamination between soil samplings, the used tools were alcohol-sterilized each time (with the ethanol burnt off). Each soil composite sample got divided into three sub-samples. The first subsample was immediately placed inside an ice box for transport to the laboratory, where it was stored at – 80 °C until its analysis. The second subsample was used to determine the soil water content (SWC); specifically, the fresh soil was placed in an aluminum box, weighed, and then oven-dried at the 120 °C for 48 h in the laboratory. The third subsample was air-dried and used to determine the soil pH, total soil carbon (TSC), total soil nitrogen (TSN), total soil phosphorus (TSP), soil available nitrogen (SAN), and soil available phosphorus (SAP).
- Measurement of Plant Traits and Calculation of Functional Diversity
The in situ photosynthetic rates (Pn) of dominant species was measured in mid-August in 2020, on sunny days between 9:30 and 11:30, using the LI-6400 Portable Photosyn- thetic System (LI-COR, Lincoln, NE, USA). For this, mature and fully expanded leaves of 49 dominant herbaceous across the species present in quadrats, 36 forbs, 5 legumes, 2 sedges, and 6 grasses, spanning 16 families (Table A1) were selected and measured using at least 10 replicate plants from across the 100-m2 plot, under an ambient CO2 concentration of 400 µmol mol-1 at 20 - 25 °C. Photosynthetic photon flux density (PPFD) was set at 1500 µmol quanta m-2 s-1, by using a red-blue 6400-02B LED light source (LI-COR INC., USA). All selected leaves were allowed to acclimate to those conditions in the chamber before taking any measurements. Three repeated values were recorded per leaf per plant until all parameters remained stable; mean values per species were calculated for use in the formal analysis. Plant community functional composition was assessed by three functional diversity indices that was FRic, FEve and FDis, respectively [54,55]. CWM weighted average of species abundance based on population functional traits to obtain community-weighted trait means [56]. Pn was included in the calculation of plant community functional composition. After the measuring the species’ photosynthetic rate, all leaves were collected to determine their SLA), LNC, LPC, and N:P. The fresh leaves were imaged by an Epson Expression 10000 XL desktop scanner. After determining the respective leaf area of the scanned samples, using WinFOLIA software (Regent Instruments Inc., Quebec City, QC, Canada), they were oven-dried at the 75 °C for 48 h and weighed. Next, the same leaf samples were digested in a mixture of sulphuric acid and hydrogen peroxide, and their LNC and LPC quantified with a SmartChem® 140 Discrete Analyzer (WESTCO Scientific
Instruments Inc., Italy).
2.4. Measurement of Soil Properties
Several soil parameters were measured according to standard protocols [47]. From
each first subsample, 5 g of frozen soil was used to determine SAN (NH4+-N and NO3-N) with the SmartChem® 140 Discrete Analyzer, after extraction with 2 M KCl for 30 min at 25 °C. To determine SAP, it was extracted with 0.5 M NaHCO3 in a 1:5 ratio (w/v) and analyzed by a UV-visible spectrophotometer (UV-2550; SHIMADZU Corp., Japan). To determine soil pH, 5 g of air-dried soil was used in slurry ratio of 1:2.5 (soil : CO2-free deionized water; w/v). For TSN, it was determined using 0.5-g air-dried soils, with a 5-mL concentrated sulfuric acid oxidation elimination done at 370 °C for 4 h, using the SmartChem® 140 Discrete Analyzer. The TSP was determined to use 0.1-g air-dried soils successively added 5-mL concentrated sulfuric acid and 10 drops interval three times perchloric acid elimination at 370 °C for distil clarify solution by SmartChem® 140 Discrete Analyzer. TSC was determined using 20 mg of air-dried soil via fast combustion at high temperature, by a TOC elemental analyzer of Germany. Soil water content was determined gravimetrically, with 10 - g fresh soil placed in an aluminum dish that was oven-dried for 48 h at 105 ± 1 °C.
2.5. Measurement of AM Fungal Biomass Soil
AM fungal biomass was quantified using the phospholipid fatty acid (PLFA) analysis, which is an efficient way to detect soil microbial biomass across different kingdoms, this based on the methodology of Bligh and Dyer (1959) as described by Bardgett et al. (1996) [57,58]. Following Bossio and Scow (1998), PLFA concentrations were determined using 8-g frozen soils to extract the lipids accordingly [59]. Fatty acid methyl esters were separated, quantified, and identified using capillary gas chromatography, with the PLFA analysis conducted on an Agilent 6890 gas chromatograph (Agilent Technologies, USA). The individual FAs were identified according to the MIDI Sherlock Microbial Identification System (MIDI Inc., USA) and each quantified, using FAME 19:0 (Matreya Inc., USA) as an internal standard. The obtained 16:1ω5c FA concentrations served as indicators of AM fungal biomass [60].
2.6. Soil DNA Extraction, PCR, and MiSeq Sequencing
Soil DNA was extracted from frozen soils (each 0.5 g) with the Magnetic Soil and Stool DNA Kit (Tiangen Biotech, Beijing, China), according to the manufacturer’s instructions. The fungi-specific primer sets AML1/AML2 and AMV4.5NF/AMDGR were used to amplify the AM fungal sequences [61,62]. The PCR amplification and library preparation are described in detail in the Supplementary Information section. The ensuing amplicons were sequenced (2 × 300-bp paired-end reads) on an Illumina MiSeq sequencer by Shanghai Biozeron Biological Technology Co. Ltd. (Shanghai, China).
2.7. Bioinformatics
After sequencing, the raw sequences were demultiplexed and quality-filtered using the FASTP pre-processing tool [38]. Next, FLASH software was used to merge the reads with the criteria described by Zheng et al. (2018) [63], after which any singleton OTUs and chimeras were removed (Mago and Salzberg, 2011) [64]. The operational taxonomic units (OTUs) were clustered at 97% similarity levels by UPARSE [65]. A representative sequence of each OTU was then selected and blasted against the AM fungal-specific MaarjAM online database [66,67]. Those sequences are specific to AM fungi and were manually picked up. Finally, these selected sequences were rarefied to the depth of the smallest sample, by using the function ‘rarefy’ and a step-size of 20 iterations (Figure A2). The AM fungal community diversity was expressed at the phylotype level, using the Shannon–Weiner index. For AM fungal community structure, in-depth analyses were performed by selecting these phylotypes with > 0.1% sequences per sample [68].
2.8. Statistical Analyses
All statistical analyses described below were performed in the R computing platform (R v4.02; R Core Team 2020). All data were log-transformed to meet parametric assumptions of normality and homogeneity of variance. Firstly, the plant traits SLA, LNC, LPC, N:P, and Pn were used to calculate functional diversity (i.e., FRic, FEve, and FDis) and CWM, by implementing the dbFD function in the ‘FD’ package for R [69]. Significant differences among the three alpine meadow sites in the response variables, including soil properties, functional diversity, community-weighed trait means, AM fungal diversity, AM fungal biomass, and the relative abundance of the top-four dominant genera, were determined by univariate one-way ANOVAs followed by Tukey’s HSD test (at an alpha level of 0.05). To examine how the AM fungal community responded to SWC, we used the Shannon-Wiener index and PLFA markers to assess its diversity and biomass, respectively [63]. To predict the variation in AM fungal diversity and biomass, those variables that differed significantly across sites were chosen for a correlation analysis. Among these, the variables found significantly correlated with AM fungal diversity or biomass were then submitted to model selection, this carried out using R package ‘MuMIn’ [70], whose dredge function automatically constructs a complete model set with all possible combinations. All candidate models were compared using the corrected Akaike Information Criterion (AICc), which is adjusted for small sample sizes; the model with the lowest AICc was deemed the best-fitting model [71]. Its determination coefficient (R2) was obtained using the function r.squaredLR in the ‘MuMIn’ package. To examine the effect of the alpine meadow site on AM fungal community structure, significant differences among the three sites were examined using the adonis function, with 999 permutations (in the ‘vegan’ package for R; likewise for the other functions). AM fungal structure was evaluated by in-depth analysis at the phylotype level, this based on Bray–Curtis distance matrix using the relative abundance ( > 0.1%) of each phylotype sequence per sample. Additionally, variables significantly affect by site were used to predict AM fungal community structure through a partial distance-based redundancy analysis (db-RDA), implemented using the function capscale. The best-fitting model was built using the function ordistep (999 permutations), and this model’s significance tested via the function anova (999 permutations).
- Results
- Responses of Soil Properties Across the Three Alpine Meadow Sites
All seven soil properties differed among the three sites (Table 1), with the pattern for SWC (F(2, 27) = 152.4, P < 0.0001) similar to their precipitation disparities (i.e., HY > MQ > HZ). In particular, TSC and TSN were markedly higher in HY than MQ and HZ, while STP, SAN, and SAP were evidently higher in HY than HZ. In contrast, soil pH was lower (acidic) in HY and MQ than HZ (Table 1).
- Responses of the AM Fungal Community Across the Three Sites
Both HY and HZ sites were co-dominated by Glomus and Acaulospora, yet only Glomus predominant in HZ (Figure 1a). We further examined the responses of the domi- nant Glomus, Acaulospora, Scutellospora and Paraglomus genera; except for Scutellospora, they all exhibited significant differences among the three sites (Figure 1b - e). Specifically, a lower SWC (soil water content) enhanced the relative abundance of Glomus significantly (Figure 1b), while that of Acaulospora as well as Paraglomus rose when going from HY to MQ and HZ (Figure 1b, e). Evidently, the AM fungal diversity changed differently among sites, median values being about two times higher in HY than HZ (Figure 2a), and so did AM fungal biomass (Figure 2b), albeit to a lesser extent, with it marginally lower in MQ than HZ. Further, as seen in Figure 3, the AM fungal community structure differed significantly among the three sites, for which the first and second axis in the db-RDA respectively explained 65.36% and 17.19% of their total variation.
Figure 1. Changes in AM fungal community composition (a), and dominant genera Glomus (b), Acaulospora (c), Scutellospora (d), and Paraglomus (e)across three alpine meadow sites on the Tibetan Plateau. The boxplots show the median (dashed white line) and quartiles; in (d) the black diamond is an outlier value. Different letters indicate significant differences between sites.
Figure 2. Changes in the AM fungal diversity (a) and biomass (b)across the three alpine meadow sites. Different letters indicate significant differences between them. The boxplots show the median (dashed white line) and quartiles.
Figure 3. A distance-based redundancy (db-RDA) analysis for the AM fungal community structure of three alpine meadow sites (HY, MQ, HZ). Total soil carbon (TSC), the plant community-weighed N:P ratio mean (N:P), community-weighed photosynthetic rate mean (Pn), and plant community functional dispersion (FDis) significantly predicted AM fungal communities, based on a stepwise model selection using permutation testing.
- Plant Community Functional Diversity and Community-Weighed Trait Means
Soil significantly altered plant community functional diversity except for FRic (Fig- ure 4a). Plant community functional diversity, when expressed as either FEve or FDis, gradually decreased from HY to MQ and HZ (Figure 4b, c), whereas FRic was not significantly different among sites (p > 0.1; Figure 4a). We next examined the community-weighed trait means, finding all of them but LNC significantly affected by site, (Figure 5a - e). Notably, in going from HY to HZ, the SLA increased by about 20% (Figure 4a) while Pn decreased about 15% (Figure 5b). In contrast, LPC was about 33% higher in MQ than HY and HZ site (Figure 5d), thus leading to a marked decrease in the N:P ratio for MQ
Figure 4. Changes in plant community functional diversity across the three alpine meadow sites, in terms of functional richness (FRic) (a), functional evenness (FEve) (b), and functional dispersion (FDis) (c). Different letters indicate significant differences between sites. The boxplots show the median (dashed white line) and quartiles; black diamonds are outlier values.
Figure 5. Changes in plant community-weighed trait means across the three alpine meadow sites, for the specific leaf area (SLA) (a), photosynthetic rate (Pn) (b), leaf nitrogen concentration (LNC) (c), the leaf phosphorus concentration (LPC) (d), and nitrogen-to-phosphorus concentration ratio (N:P) (e). Different letters indicate significant differences between sites. The boxplots show the median (dashed white line) and quartiles; the black diamond is an outlier valuein (e).
- The AM Fungal Community in Relation to Plant Community Functional Diversity, Community-Weighed Trait Means, and Soil Properties
We used correlations, model selection, and db-RDA to predict the AM fungal com- munity response across the three sites. Evidently, most soil properties were significantly correlated with AM fungal diversity, while plant community-weighed trait means for LPC and N:P were both significantly correlated with AM fungal biomass (Figure 6). Model selection demonstrated that SAP and STN together explained 47% of the variation in AM fungal diversity, while LPC alone explained 15% of the variation in AM fungal biomass (Table 2). Moreover, the db-RDA in Figure 3 revealed that TSC, N: P, Pn, and FDis significantly collectively explained 40.25% of variation in AM fungal community structure (F(4, 25) = 4.21, p = 0.001).
Figure 6. Pearson correlations between AM fungal diversity and biomass with plant community functional diversity, community-weighted trait means and soil properties. Pink indicates positively correlated, blue indicates negatively correlated. A circle’s size corresponds to the magnitude (r value) of that correlation. *, P < 0.05; **, P< 0.01; ***, P< 0.001. FRic-functional richness; FEve-functional evenness; FDis-functional dispersion. SLA-specific leaf area; LNC -leaf nitrogen concentration; LPC-leaf phosphorus concentration; N:P-leaf nitrogen and phosphorus concentration ratio; Pn- photosynthetic rate; SWC- soil water content; TSC-total soil carbon; TSN-total soil nitrogen; TSP- total soil phosphorus; SAP- soil available phosphorus; SAN-soil available nitrogen.
Table 2. Best-fitting models were selected using the function ‘dredge’ for AM fungal diversity and biomass.
- Discussion
We assessed key aspects of the AM fungal community, namely its diversity, biomass, community composition and structure, in the alpine meadow ecosystem of the Tibetan Plateau. Our results revealed remarkable changes in AM fungal diversity, plant community functional diversity, and soil nutrients across three alpine meadow sites. The higher precipitation at HY led to a higher SWC, which was associated with a substantial increase in its AM fungal diversity. This result agrees well with work by Zhang et al. (2016), who found that AM fungal diversity gradually fell as precipitation declined in the alpine steppe of the Tibetan Plateau. A recent study showed that long-term drought in an alpine steppe markedly reduced its AM fungal diversity [72], whereas a similar disturbance had negligible effects in mesic grassland [33]. Conversely, a 7-year period of increased precipitation led to greater AM fungal richness in steppe grassland of Inner Mongolia [31]. These results indicate that how AM fungal diversity responds to changes in precipitation might depend on a grassland ecosystem’s properties, especially its ambient soil nutrients. Here, our correlations and model selection demonstrated that SAP and TSN could positively predict the AM fungal diversity of alpine meadows. Our results are thus consistent with those of Liu et al. (2012) [73], who found that a combination of nitrogen (N) and phospho- rus (P) added at a low dose enhanced AM fungal diversity as well as root colonization in an alpine meadow, but reduced both features when these nutrients were applied at a high dose. Alongside our results, this suggests that AM fungi are N- and P- limited in alpine meadows.
That dual limitation should spur AM fungal taxa to compete for available N and P in soils, because AM fungal tissue requires more N and P than do typical plant tissues. This intensified competition could lead to a stark reduction in AM fungal diversity. However, since the threshold for nutrient limitation tends to be lower for AM fungi than for plants [74], low-dose N and P additions often ease AM fungal competition for N and P, leading to an increase in AM fungal diversity [75,76]. By contrast, high-dose N and P additions can offset the N and P limitation faced by various plant species [77]. Plant hosts will reduce their C allocation to AM fungi, which would allow AM fungal taxa to compete for derived-plant C, leading to less AM fungal diversity [78,79]. We find that both AM fungal diversity and biomass differed across three alpine meadow sites. Notably, AM fungal biomass is slightly higher in MQ than HZ. This could be due to between-site discrepancies in plant community functional composition, especially the relative abundance of grasses and forbs [80]. In MQ, the high accumulative relative abundance of forb specie may spur the plant community to demand more P, because the LPC is markedly higher in forbs than grasses and forbs with thick roots depend more on AM fungal hyphal networks for nutrient capture than do grasses whose roots are thin [81,82]. Hence, that the AM fungal biomass is predicted by the plant community weighted-LPC in our study could be explained by the mean of cumulative relative abundance more than 52% and higher LPC of forbs, respectively (Table A1, Figure 5d). We recorded the presence of 36 forb species, constituting 73.47% of all 49 plant species observed in the three surveyed alpine meadow sites. These forbs mostly rely on the C3 photosynthesis pathway, and more than half of them are native perennial plants typical of alpine meadow communities on the Tibetan Plateau (Table A1).
We also found that fungal members of Glomus dominated the HZ site. According to work by Egerton-Warburton et al. (2007) [75], Glomus is predominant in semiarid sites, while Scutellospora, Gigaspora, and are abundant in mesic sites of grasslands. Recently, we observed that long-term drought bolstered the relative abundance of Glomus in an alpine steppe [72]. This varied response of AM fungal taxa or clades to soil moisture levels may partly arise from their life history strategies based on functional traits [83]. It is known that certain AM fungal taxa or clades differ in their C demands, C-use efficiency, and hyphae-related traits [84]. For example, members of Glomus are more effective at C utilization; so, despite less available C, they could maintain their hyphal growth and sporulation [85].
Long-term drought often depresses photosynthetic activity in the canopy of plants and reduces their C allocation to belowground to host AM fungi [86], possibly favoring the predominance of Glomus. The C demands of distinct AM fungal taxa are also thought to be trait-linked [83]. For example, those AM fungal taxa with delicate hyphae and small spores require less C [84]. Although a trait-based framework has been theoretically proposed to explain the responses of AM fungal clades to environmental changes [87,88], empirically tracking in situ the C allocated to differing AM fungal taxa and evaluating individual AM fungal traits are both daunting tasks. At this point, we can only surmise that, with climatic changes, AM fungal communities maximize their own fitness through coordinated shifts in their composition [89].
Another important finding of our study was that variation in AM fungal community structure is predicted well by plant community functional composition as well as soil properties. Several studies have shown that changed precipitation regimes can alter AM fungal community structure, but not AM fungal diversity [30,33]. This would suggest a pivotal role of AM fungal community structure that is site-responsive, and the changes in AM fungal community structure may have important consequences for ecosystems long before AM fungal diversity is threatened by extinction. Accordingly, it is crucial to adequately consider AM fungal community structure for grassland management, conservation, and restoration in the face of ongoing climate change.
We find that plant traits involved in the C-nutrient economy can predict the variation in AM fungal community structure. The best predictors were the plant community-weighed N:P ratio and Pn, along with the TSC. In previous models only the effects of soil N:P ratio on AM fungal communities were predicted [78,79]. Leaf N:P ratios may be used to gauge the degree of plant nutrient availability and limitation, and they play an essential role in photosynthesis processes [90,91]. Few studies, however, have specifically focused on how the leaf N:P ratio at the plant community level may affect the AM fungal community [77]. Due to their ramified hyphae, AM fungi can transfer P outside the depletion zone and some AM fungal taxa can directly uptake organic P [92]. In contrast, AM fungal hyphae may not provide an N-uptake advantage over roots due to the greater mobility of nitrate and ammonium in soils [78]. Yet a recent meta-analysis did uncover a positive effect of AM fungi on rates of N uptake [20]. Several lines of evidence have clearly demonstrated that AM fungal C availability is able to stimulate both N and P uptake and transport in the hyphal network [27,93]. Therefore, plant traits critically involved in nutrient-C exchange via AM fungal symbionts may figure prominently in the community assembly of AM fungi.
(suggested new paragraph) As a major contributor to soil organic C, photosynthesis directly provides C for AM fungal growth and development [29]. Indeed, the AM fungi-derived C from plants constitutes, on its own, an important fraction of TSC [37]. Moreover, AM fungal hyphae also exude various C compounds into soils, such as sugars, carboxylates, and amino acids [94]. We also observed that FDis is a significant predictor of AM fungal community structure in the best-fitting model. The FDis reflects the degree of trait dispersion within a plant community, whereby a high FDis augments overall resource utilization (e.g., of water, light, and nutrients) via trait differentiation [12], and this conceivably may further affect the community structure of AM fungi. The number of these plant species associated with soil nutrients declined sequentially from HY to MQ, and finally to HZ (Table A1; Table 1). Hence, our results provide compelling evidence that plant functional composition and soil properties act together to shape the AM fungal community of alpine meadow on the Tibetan Plateau.
- Conclusions
Our study shows that the diversity of AM fungi, as well as their community compo- sition and structure, can vary markedly across alpine meadow sites, while their biomass tends to remain stable across space. The high SWC induced by more precipitation is capable of maintaining greater plant functional diversity, leading to changed soil nutrients and AM fungal diversity, whereas Glomus prevails in site conditions of low SWC and low soil nutrient availability. Potential differences in traits of AM fungi or their life history strategies could impact the responses of AM fungal community composition in alpine meadow. This study highlights the importance of considering both plant traits and soil properties in tandem, particularly those most closely associated with nutrient-C exchange via AM fungal symbionts, in shaping AM fungal communities in the alpine meadow ecosystem of the Tibetan Plateau. Our findings have valuable implications for further understanding the response of AM fungi and their diversity to nutrient-C exchange dynamics tied to plant traits and soil properties in grasslands.
Supplementary Materials: The following supporting information can be downloaded at:
www.mdpi.com/xxx/s1, PCR amplification; Figure A1: Map showing the location of the three alpine meadow sites (Hongyuan, Maqu, and Hezuo) sampled on the Qinghai-Tibetan Plateau, China. Fig- ure A2. Rarefaction curves showing the AM fungal phylotype abundances across all samples in the three alpine meadow sites of this study. Table A1: Description of the 49 herbaceous species investi- gated in the three alpine meadow sites.
Author Contributions: Conceptualization, S.Z. and Z.Y.; methodology, S.Z.; software, Z.Y.; valida- tion, S.Z., Z.Y. and X.Y.; formal analysis, S.Z. and Z.Y.; investigation, X.Y., X.M. and Q.M; resources, S.Z. and M.M.; data curation, S.Z.; writing—original draft preparation, S.Z. and Z.Y.; writing—re- view and editing, S.Z. and M.M.; visualization, Z.Y. and J.Z.; supervision, S.Z.; project administra- tion, S.Z.; funding acquisition, S.Z. All authors have read and agreed to the published version of the manuscript.
Funding: This study was supported by National Natural Science Foundation of China (32260298 and 31660160), Ecological Civilization Construction Key Research and Development Special Project of Gansu Province (24YFFA061).
Institutional Review Board Statement: Not applicable.
Informed Consent Statement: Not applicable.
Data Availability Statement: All available data are published in this paper and the Supplementary
Acknowledgments: We thank ZZ for providing help in field sampling and for insightful sugges- tions that contributed to improvements in the manuscript.
Conflicts of Interest: The authors declare no conflicts of interest.
( Note: Journal requirements for References are presented at https://www.mdpi.com/journal/jof/instructions
References should be described as follows, depending on the type of work:
- Journal Articles:
1. Author 1, A.B.; Author 2, C.D. Title of the article. Abbreviated Journal Name Year, Volume, page range. - Books and Book Chapters:
2. Author 1, A.; Author 2, B. Book Title, 3rd ed.; Publisher: Publisher Location, Country, Year; pp. 154–196.
3. Author 1, A.; Author 2, B. Title of the chapter. In Book Title, 2nd ed.; Editor 1, A., Editor 2, B., Eds.; Publisher: Publisher Location, Country, Year; Volume 3, pp. 154–196.
References
Hanasaki, N.; Fujimori, S.; Yamamoto, T.; Yoshikawa, S.; Masaki, Y.; Kainuma, M.; Kanamori, Y.; Masui, T.; Takahashi, K.; Kanae, S.A global water scarcity assessment under Shared Socio-economic Pathways 49 - Part 2: Water availability and scarcity. Hydrol. Earth Syst. Sc. 2013, 17, 2375-2391.
Chen, H.; Sun, J.; Chen, X. Projection and uncertainty analysis of global precipitation-related 30 extremes using CMIP5 models. Int. J. Climatol. 2014, 34, 2730-2748.
Westra, S.L.V.; Alexander, L.V.; Zwiers, F.W. Global increasing trends in annual maximum daily precipitation. J. Climate 2013, 26, 3904-3918.
Sun, Q.; Zhang, X.; Zwiers, F.; Westra, S.; Alexander, L. A global, continental and regional analysis of changes in extreme precipitation. J. Climate 2021, 34, 243-258.
Knapp, A.K.; Hoover, D.L.; Wilcox, K.R.; Avolio, M.L.; Koerner, S.E.; La Pierre, K.J.; Loik, M.E.; Luo, Y.; Sala, O.E.; Smith, M.D. Characterizing differences in precipitation regimes of extreme wet and dry years: implications for climate change experiments. Glob. Change Biol. 2015, 21, 2624-2633.
Grime, J.P.; Brown, V.K.; Thompson, K.; Master, G.J.; Hilier, S.H.; Clark, I.P.; Askew, A.P.; Corker, D.; Kielty, J.P. The response of two contrasting limestone grasslands to simulated climate change. Science 2000, 289, 762-765.
Yang, H.; Wu, M.; Liu, W.; Zhang, Z.; Zhang, N.; Wan, S. Community structure and composition in response to climate change in a temperate steppe. Global Change Biol. 2011, 17, 452-465.
Liu, H.; Mi, Z.; Lin, L.; Wang, Y.; Zhang, Z.; Zhang, F.; Wang, T.; Liu, L.; Zhu, B.; Cao, G.; Zhao, X.; Sanders, N.J.; Classen, A.; Reich, P.B.; He, J. Shifting plant species composition in response to climate change stabilizes grassland primary production. Proc. Natl. Acad. Sci. USA 2018,115, 4051-4056.
Griffin-Nolan, R.J.; Blumenthal, D.M.; Collins, S.L.; Farkas, T.E.; Hoffman, A.M.; Mueller, K.E.; Ocheltree, T.W.; Smith, M.D.; Whitney, K.D.; Knapp, A.K. Shifts in plant functional composition following long-term drought in grasslands. J. Ecol. 2019, 107, 2133-2148.
Luo, W.; Griffin-Nolan, R.J.; Ma, W.; Liu, B.; Zuo, X.; Xu, C.;Yu, Q.; Luo, Y.; Mariotte, P.; Smith, M.D.; Collins S.L.; Knapp, A.K.; Wang, Z.; Han, X. Plant traits and soil fertility mediate productivity losses under extreme drought in C3 grasslands. Ecology 2021, 102, e03465.
Zuo, X.; Zhao, S.; Cheng, S.; Hu, Y.; Wang, S.; Yue, P.; Liu, R.; Knapp, A.K.; Smith, M.D.; Yu, Q.; Koerner, S. Functional diversity response to geographic and experimental precipitation gradients varies with plant community type. Funct. Ecol. 2021, 35, 2119- 5162132.
Gross, N.; Suding, K.N.; Lavorel, S.; Roumet, C. Complementarity as a mechanism of coexistence betwee n functional groups of grasses. J. Ecol. 2007, 95, 1296-1305.
Valencia, E., Maestre, F.T.; Le Bagousse-Pinguet, Y.; Quero, J.L.; Tamme, R.; Borger, L.; García-Gómez, M.; Gross, N. Functional diversity enhances the resistance of ecosystem muntifunctionality to acridity in Mediterranean drylands. New Phytol. 2015, 206, 521
660-671.
Gross, N.; Robson, T.M.; Lavorel, S.; Albert, C.; Le Bagousse-Pinguet, Y.; Guillemin, R. Plant response traits mediate the effects 523 of subalpine grasslands on soil moisture. New Phytol. 2008, 180, 652-662.
Faucon, M.P.; Houben, D.; Lambers, H. Plant functional traits: soil and ecosystem services. Trends Plant Sci. 2017, 22, 385-394.
Brundrett, M.C.; Tedersoo, L. Evolutionary history of mycorrhizal symbioses and global host plant diversity. New Phytol. 2018, 220, 1108-1115.
Soudzilovskaia, N.A.; van Bodegom, P.M.; Terrer, C.; van’t Zelfde, M.; McCallum, I.; Luke McCormack, M.; Fisher, J.B.; Brundrett, M.C.; de Sá, N.C.; Tedersoo, L.Global mycorrhizal plant distribution linked to terrestrial carbon stocks. Nat. Commun. 2019, 10, 5077.
Hawkins, H.J.; Cargill, R.I.M.; Van Nuland, M.E.; Hagen, S.C.; Field, K.J.; Sheldrake, M.; Soudzilovskaia, N.A.; Kiers, E.T. Mycorrhizal mycelium as a global carbon pool. Curr. Biol. 2023, 33, PR560-R573.
Smith, S.E.; Facelli, E.; Pope, S.; Andrew Smith, F. Plant performance in stressful environments: interpreting new and established knowledge of the roles of arbuscular mycorrhizas. Plant and Soil 2010, 326, 3-20.
- Delavaux, C.; Smtth-Ramesh, L.M.; Kuebbng, S.E. Beyond nutrients: a meta-analysis of the diverse effects of arbuscular mycorrhizal fungi on plants and soils. Ecology 2017, 98, 2111-2119.
Tedersoo, L.; Bahram, M.; Zobel, M. How mycorrhizal associations drive plant population and community biology. Science 2020, 367, eaba1223. DOI: 10.1126/science.aba1223.
Davison, J.; Moora, M.; Öpik, M.; Adholeya, M.; Ainsaar, L.; Bâ, A.; Diedhiou, A.G.; Hiiesalu, I.; Jairus, T.; Johnson, N.C.; Kane, A.; Koorem, K.; Kochar, M.; Ndiaye, C.; Pärtel, M.; Reier, Ü; Saks, Ü.; Singh, R.; Zobel, M. Global assessment of arbuscular mycorrhizal fungus diversity reveals very low endemism. Science 2015, 349, 970-973.
Wu S.; Fu W., Rillig M.C.; Chen B.; Zhu Y.; Huang L.Soil organic matter dynamics mediated by arbuscular mycorrhizal fungi— An updated conceptual framework. New Phytol. 2023, ????volume, pages.???? DOI: 10.1111/nph.19178.
Kaschuk, G.; Kuyper, T.W.; Leffelaar, P.A.; Hungria, M.; Giller, K.E. Are the rates of photosynthesis stimulated by the carbon sink strength of rhizobial and arbuscular mycorrhizal symbioses? Soil Biol. Biochem. 2009, 41, 1233-1244.
Schweigert, M.; Herrmann, S.; Miltner, A.; Fester, T.; Kästner, M. Fate of ectomycorrhizal fungal biomass in a soil bioreactor system and its contribution to soil organic matter formation. Soil Biol. Biochem. 2015, 88, 120-127.
Gavito, M.E.; Jakobsen, I.; Mikkelsen, T.N.; Mora, F. Direct evidence for modulation of photosynthesis by an arbuscular mycorrhiza-induced carbon sink strength. New Phytol. 2019, 223, 896-907.
Bücking, H.; Shachar-Hill, Y. Phosphate uptake, transport and transfer by the arbuscular mycorrhizal fungus Glomus intraradices is stimulated by increased carbohydrate availability. New Phytol. 2005, 165, 899-912.
Kakouridis, A.; Hagen, J.A.;Kan, M.P.;Feldman, L.J.;Herman, D.J.;Weber, P.K.; Pett-Ridge, J.;Firestone, M.K.Routes to roots: direct evidence of water transport by arbuscular mycorrhizal fungi to host plants. New Phytol. 2022, 236, 210-221.
Smith, S.E.;Read, D.J. Mycorrhizal Symbiosis. Academic Press, Cambridge, Massachusetts USA 2008.
Chen, Y.; Xu, Z.; Xu, T.; Veresoglou, S.; Yang, G.; Chen, B. Nitrogen deposition and precipitation induced phylogenetic clustering of arbuscular mycorrhizal fungal communities. Soil Biol. Biochem. 2017, 115, 233-242.
Li, X., Zhu, T.; Peng, F.; Chen, Q.; Lin, S.; Christie, P.; Zhang, J. Inner mongolian steppe arbuscular mycorrhizal fungal communities respond more strongly to water availability than to nitrogen fertilization. Environ. Microbiol. 2015, 17, 3051-3068.
House, G.L.; Bever, J.D. Disturbance reduces the differentiation of mycorrhizal fungal communities in grasslands along a precipitation gradient. Ecol. App. 2018, 28, 736-748.
Deveautour, C.; Donn S.; Power, S.A.; Bennett, A.E.; Powell, J.R. Experimentally altered precipitation regimes and host root traits affect grassland arbuscular mycorrhizal fungal communities. Mol. Ecol. 2018, 27, 2152-2163.
Deveautour, C.; Power, S.; Barnett, K.; Ochoa-Hueso, R.; Powell, J. Temporal dynamics of mycorrhizal fungal communities and co-associations with grassland plant communities following experimental manipulation of precipitation. J. Ecol. 2019, 108, 515- 527. 565
Staddon, P.L.; Thompso, K.; Jakobsen, I.; Grime, J.P.; Askew, A.P.; Fitter, A.H. Mycorrhizal fungal abundance is affected by long-term climatic manipulations in the field. Glob. Change Biol. 2003, 9, 186-194.
Johnson, N.C.; Rowland, D.L.; Corkidi, L.; Egerton-Warburton, L.M.; Allen, E.B. Nitrogen enrichment alters mycorrhizal allocation at five mesic to semiarid grasslands. Ecology 2003, 84, 1895-1908.
Rillig, M.C. Arbuscular mycorrhizae and terrestrial ecosystem processes. Ecol. Lett. 2004, 7, 740-754.
Chen, H.; Zhu, Q.; Peng, C.; Wu, L.; Wang, Y.; Fang, X.; Wu, J. The impacts of climate change and human activities on biogeochemical cycles on the Qinghai-Tibetan Plateau. Glob. Change Biol. 2013, 19, 2940-2955.
Ding, D.; Yang, T.; Zhao, Y.; Liu, D.; Wang, X.; Yao, Y.; Piao, S.. Increasingly important role of atmospheric aridity on Tibetan alpine grasslands. Geophys. Res. Lett. 2018, 45, 2852-2859.
He, J.; Dong, S.; Shang, Z.; Sundqvist, M.K.; Wu, G.; Yang, Y. Above-belowground interactions in alpine ecosystems on the roof of the world. Plant and Soil 2021, 458, 1-6.
Gai, J.P.; Feng, G.; Cai, X.B.; Christie, P.; Li, X.L. A preliminary survey of the arbuscular mycorrhizal status of grassland plants in southern Tibet. Mycorrhiza 2006, 16, 191-196.
van der Heijden, M.G.A.; Klironomos, J.N.; Ursic, M.; Moutoglis, P.; Streitwolf-Engel, R.; Boller, T.; Wiemken, A.; Sanders, I.R. Mycorrhizal fungal diversity determines plant biodiversity, ecosystem variability and productivity. Nature 1998, 396, 69-72.
Collins, C.D.; Foster, B.L. Community-level consequences of mycorrhizae depend on phosphorus availability. Ecology 2009, 90, 2567-2576. 582
Bulgarelli, R.G.; Marcos, F.C.C.; Ribeiro, R.V.; de Andrade, S.A.L.Mycorrhizae enhance nitrogen fixation and photosynthesis in phosphorus starved soybean (L. Merrill). Environ. Exp. Bot. 2017, 140, 26-33.
Zhu, X.; Song, F.;Xu, H. Arbuscular mycorrhizae improves low temperature stress in maize via alterations in host water status and photosynthesis. Plant and Soil. 2010, 331, 129-137.
Liu, Y.; Reich, P.B.; Li, G.; Sun, S.Shifting phenology and abundance under experimental warming alters trophic relationships and plant reproductive capacity. Ecology 2011, 92, 1201-1207.
Zhang, S.; Chen, D.; Sun, D.; Wang, X.; Smith, J.; Du, G. Impacts of altitude and position on the rates of soil nitrogen mineralization and nitrification in alpine meadows on the eastern Qinghai-Tibetan Plateau, China. Biol. Fert. Soils 2012, 48, 393-400.
Yang, Z.L.; Li, J.Y.; Xiao, R.; Zhang, C.H.; Ma, X.J.; Du, G.Z.; Li, G.Y.; Jiang, L. Losses of low-germinating, slow-growing species prevent grassland composition recovery from nutrient amendment. Global Change Biol. 2024, 30, e17264.
An, H.; Zhao, Y.; Ma, M. Precipitation controls seed bank size and its role in alpine meadow community regeneration with increasing altitude. Glob. Change Biol. 2020, 26, 5767-5777.
Xiao. Y.; Liu, X.; Zhang, L.; Song, Z.; Zhou, S. The allometry of plant height explains species loss under nitrogen addition. Ecol. Lett. 2021, 24, 553-562. 596
Cheng, Y.; Rutten, G.; Liu, X.; Ma, M.; Song, Z.; Maaroufi, N.I.; Zhou, S. Host plant height explains the effect of nitrogen enrichment on arbuscular mycorrhizal fungal communities. New Phytol. 2023, 240, 399-411.
Liu, J.; Zhang, X.; Song, F.; Zhou, S.; Cadotte, M.W.; Bradshaw, C.J. Explaining maximum variation in productivity requires phylogenetic diversity and single functional traits. Ecology 2015, 96, 176-183.
Niu, K.;Choler, P.;Zhao,B.;Du, G.The allometry of reproductive biomass in response to land use in Tibetan alpine grass- lands.Funct. Ecol. 2009, 23, 274-283.
Mason, N.W.H.; Mouillot, D; Lee, W.G.; Wilson, J.B.Functional richness, functional evenness and functional divergence: the primary components of functional diversity. Oikos 2005, 111, 112-118.
Mouillot, D.; Mason, W.H.N.; Dumay, O.; Wilson, J.B. Functional regularity: a neglected aspect of functional diversity. Oecologia 2005, 142, 353-359.
Garnier, E.; Cortez, J.; Billes, G.; Navas, M.L.; Roumet, C.; Debussche, M.; Laurent, G.; Blanchard, A.; Aubry, D.; Bellmann, A.;Neill, C.;Toussaint, J.P.Plant functional markers capture ecosystem properties during secondary succession. Ecology 2004, 85, 2630-2637.
Bligh, E.G.; Dyer, W.J.A rapid method of total lipid extraction and purification. Can. J. Biochem. Physiol. 1959, 37, 911-917.
Bardgett, R.D., Hobbs, P.J., Frostegard, A. Changes in soil fungal:bacterial biomass ratios following reductions in the intensity of management of an upland grassland. Biol. Fertil Soils 1996, 22, 261-264.
Bossio, D.A.; Scow, K.M. Impacts of carbon and flooding on soil microbial communities: phospholipid fatty acid profiles and substrate utilization patterns. Microb. Ecol. 1998, 35, 265-278.
Olsson, P.A.; Baath, E.; Jakobsen, I.; Soderstrom, B. The use of phospholipid and neutral lipid fatty acids to estimate biomass of arbuscular mycorrhizal fungi in soil. Mycol. Res. 1995,99, 623-629.
Lee, J.; Lee, S.; Young, J.P.W. Improved PCR primers for the detection and identification of arbuscular mycorrhizal fungi. FEMS-Microbiol. Ecol. 2008, 65, 339-349.
Sato, K.; Suyama, Y.; Saito, M.; Sugawara, K. A new primer for discrimination of arbuscular mycorrhizal fungi with polymerase chain reaction-denature gradient gel electrophoresis. Grassl Sci. 2005, 51, 179-181.
Zheng, Z.; Ma, P.; Li, J.; Ren, L.; Bai, W.; Tian, Q.; Sun, W.; Zhang, W.Arbuscular mycorrhizal fungal communities associated with two dominant species differ in their responses to long-term nitrogen addition in temperate grasslands. Funct. Ecol. 2018, 32, 1575-1588.
Magoč, T.; Salzberg, S.L.FLASH: fast length adjustment of short reads to improve genome assemblies. Bioinformatics 2011,27, 2957-2963.
Edgar, R.C. UPARSE: Highly accurate OTU sequences from microbial amplicon reads. Nat. Methods 2013, 10, 996-998.
Öpik, M.; Metsis, M.; Daniell, T.J.; Zobel, M.; Moora, M.Large-scale parallel 454 sequencing reveals ecological group specificity of arbuscular mycorrhizal fungi in a boreonemoral forest. New Phytol. 2009, 184, 424-437.
Öpik, M.;Vanatoa, A.; Vanatoa, E.;Moora, M.;Davison, J.;Kalwij, M.J.;Reier, Ü.;Zobel, M.The online database MaarjAM reveals global and ecosystemic distribution patterns in arbuscular mycorrhizal fungi (Glomeromycota). New Phytol. 2010, 18, 223-241.
Harman, K.; van der Heijden, M.G.A.; Wittwer, R.A.; Banerjee, S.; Walser, J.C.; Schlaeppi, K. Cropping practices manipulate abundance patterns of root and soil microbiome members paving the way to smart farming. Microbiome2018,6, 1-14. Laliberte, E.; Legendre, P. A distance-based framework for measuring functional diversity from multiple traits. Ecology 2010, 91,
- Barton, K. MuMIn: Multi-model inference. R package version 1.15.6. Retrieved from https://CRAN.R-project.org/pack- age=MuMIn. 2016, 71. 72.
Burnham, K.P.; Anderson, D.R. Model selection and multimodel inference, 2nd ed. Springer-Verlag. 2002.
Zheng, Z.; Ma, X.; Zhang, Y.; Liu, Y.; Zhang, S. Soil properties and plant community-level traits mediate arbuscular mycorrhizal fungal response to nitrogen enrichment and altered precipitation. Appl. Soil Ecol.2022,169, 104246. 73. Liu, Y.; Shi, G.; Mao, L.; Cheng, G.; Jiang, S.; Ma, X.; An, L.; Du, G.; Johnson, N.C.; Feng, H. Direct and indirect influences of 8 yr of nitrogen and phosphorus fertilization on Glomeromycota in an alpine meadow ecosystem. New Phytol. 2012, 194, 523-535.
- Treseder, K.K.; Allen, M.F. Direct nitrogen and phosphorus limitation of arbuscular mycorrhizal fungi: a model and field test. New Phytol. 2002, 155, 507-515. 643
- Egerton-Warburton, L.M.; Johnson, N.C.; Allen, E.B. Mycorrhizal community dynamics following nitrogen fertilization: a cross- site test in five grasslands. Ecol. Monogr. 2007, 77, 527-544.
- Porras-Alfaro, A.; Herrera, J.; Natvig, D.O.; Sinsabaugh, R.L. Effect of long-term nitrogen fertilization on mycorrhizal fungi associated with a dominant grass in a semiarid grassland. Plant and Soil. 2007, 296, 65-75.
- Johnson, N.C.; Rowland, D.L.; Corkidi, L.; Allen, E.B. Plant winners and losers during grassland N-eutrophication differ in biomass allocation and mycorrhizas. Ecology 2008,89, 2868-2878.
- Johnson, N.C.Resource stoichiometry elucidates the structure and function of arbuscular mycorrhizas across scales. New Phytol. 2010, 135, 575-585. 651
- Grman, E.;Robinson, T.M.P. Resource availability and imbalance affect plant-mycorrhizal interactions: a field test of three hypotheses. Ecology 2013, 94, 62-71. 653
- Hetrick, B.A.D.; Wilson, G.W.T.; Todd, T.C. Relationships of mycorrhizal symbiosis, rooting strategy, and phenology. Can. J. Bot. 1992,70, 1521-1528. 655
- Tian, Q.; Lu, P.; Ma, P.; Zhou, H.; Yang, M.; Zhai, X.; Chen, M.;Wang, H.; Li, W.; Bai, W.; Lambers, H.; Zhang, W.H. Processes at the soil-root interface determine the different responses of nutrient limitation and metal toxicity in forbs and grasses to nitro- gen enrichment. J. Ecol. 2021,109, 927-938.
- Ma, Z.Q.; Guo, D.L.; Xu, X.L.; Lu, M.Z.; Bardgett, R.D.; Eissenstat, D.M.; McCormack, M.L.; Hedin, L.O. Evolutionary history resolves global organization of root functional traits. Nature 2018,555, 94-97.
- Chagnon, P.L.;Bradley, R.L.; Maherali, H.;Klironomos, J.N.A trait-based framework to understand life history of mycorrhizal fungi. Trends Plant Sci. 2013a.18, 484-491.
- Hart, M.M.; Reader, R.J. Taxonomic basis for variation in the colonization strategy of arbuscular mycorrhizal fungi. New Phytol. 2002,153, 335-344. 664
- Douds, D.D.; Schenck, N.C. Relationship of colonization and sporulation by VA mycorrhizal fungi to plant nutrient and carbo- hydrate contents. New Phytol. 1990, 116, 621-627.
- Fuchslueger, L.; Bahn, M.; Fritz, K.; Hasibeder, R.; Richter, A. Experimental drought reduces the transfer of recently fixed plant carbon to soil microbes and alters the bacterial community composition in a mountain meadow. New Phytol. 2014, 201, 916-927.
- Treseder, K.K.; Allen, E.B.; Egerton-Warburton, L.M.; Hart, M.M.; Klironomos, J.N.; Maherali, H.;Tedersoo, L. Arbuscular mycorrhizal fungi as mediators of ecosystem responses to nitrogen deposition: A trait-based predictive framework. J. Ecol. 2018, 106, 480-489.
- Dueñas, J.F.; Camenzind, T.; Roy, J.; Hempel, S.; Homeier, J.; Suárez, J.P.; Rillig, M.C. Moderate phosphorus additions consist- ently affect community composition of arbuscular mycorrhizal fungi in tropical montane forests in southern Ecuador. New Phytol. 2020, 227, 1505-1518.
- Chagnon, P.L.;Bradley, R.L. Evidence that soil nutrient stoichiometry controls the competitive abilities of arbuscular mycorrhizal vs. root-borne non-mycorrhizal fungi. Fungal Ecol. 2013b, 6, 557-560.
- Reich, P.B.; Oleksyn, J. Global patterns of plant leaf N and P in relation to temperature and latitude. Proc. Natl. Acad. Sci. USA 2004, 101, 11001-11006.
- Schreeg, L.A.; Santiago, L.S.; Wright, S.J.; Turner, B.L. Stem, root, and older leaf N:P ratios are more responsive indicators of soil nutrient availability than new foliage. Ecology 2014, 95, 2062-2068.
- Lambers, H.;Bishop, J.G.;Hopper, S.D.;Laliberte, E.;Zuniga-Feest, A. Phosphorus-mobilization ecosystem engineering: the roles of cluster roots and carboxylate exudation in young P-limited ecosystems. Ann. Bot. 2012,110, 959-968.
- 94. Fellbaum, C.R.; Gachomo, E.W.; Beesetty, Y.; Choudhari, S.; Strahan, G.D.; Pfeffer, P.E.; Kiers, E.T.; Buecking, H. Carbon avail- ability triggers fungal nitrogen uptake and transport in arbuscular mycorrhizal symbiosis.Proc. Natl. Acad. Sci. USA 2012, 109, 2666-2671.
Zhang, J.; Wang, F.; Che, R.; Wang, P.; Liu, H.; Ji, B.; Cui, X. Precipitation shapes communities of arbuscular mycorrhizal fungi in Tibetan alpine steppe. Sci. Rep. 2016, 6, 23488.